# Prevalence of *arps10*, *fd*, *pfmdr-2*, *pfcrt* and *pfkelch13* gene mutations in *Plasmodium falciparum* parasite population in Uganda

**Moses Ocan** [1]*, **Fred Katabazi Ashaba**[2], **Savannah Mwesigwa**[3], **Kigozi Edgar**[2], **Moses R. Kamya**[4,5], **Sam L. Nsobya**[4,6]

**1** Department of Pharmacology & Therapeutics, Makerere University, Kampala, Uganda, **2** Makerere University Biomedical Research Center, Kampala, Uganda, **3** Department of Immunology and Molecular Biology, Makerere University, Kampala, Uganda, **4** Infectious Disease Research Collaboration (IDRC), Kampala, Uganda, **5** Department of Medicine, Makerere University, Kampala, Uganda, **6** Department of Pathology, Kampala, Uganda

* ocanmoses@gmail.com

**Data Availability Statement:** All relevant data are within the manuscript and its Supporting information files.

## Abstract

In Uganda, Artemether-Lumefantrine and Artesunate are recommended for uncomplicated and severe malaria respectively, but are currently threatened by parasite resistance. Genetic and epigenetic factors play a role in predisposing *Plasmodium falciparum* parasites to acquiring *Pfkelch13* (K13) mutations associated with delayed artemisinin parasite clearance as reported in Southeast Asia. In this study, we report on the prevalence of mutations in the K13, *pfmdr-2* (*P. falciparum* multidrug resistance protein 2), *fd* (ferredoxin), *pfcrt* (*P. falciparum* chloroquine resistance transporter), and *arps10* (apicoplast ribosomal protein S10) genes in *Plasmodium falciparum* parasites prior to (2005) and after (2013) introduction of artemisinin combination therapies for malaria treatment in Uganda. A total of 200 *P. falciparum* parasite DNA samples were screened. Parasite DNA was extracted using QIAamp DNA mini kit (Qiagen, GmbH, Germany) procedure. The PCR products were sequenced using Sanger dideoxy sequencing method. Of the 200 *P. falciparum* DNA samples screened, sequencing for mutations in *K13*, *pfmdr-2*, *fd*, *pfcrt*, *arps10* genes was successful in 142, 186, 141, 128 and 74 samples respectively. Overall, we detected six (4.2%, 6/142; 95%CI: 1.4–7.0) K13 single nucleotide polymorphisms (SNPs), of which 3.9% (2/51), 4.4% (4/91) occurred in 2005 and 2013 samples respectively. All four K13 SNPs in 2013 samples were non-synonymous (A578S, E596V, S600C and E643K) while of the two SNPs in 2005 samples, one (Y588N) is non-synonymous and the other (I587I) is synonymous. There was no statistically significant difference in the prevalence of K13 ($p = 0.112$) SNPs in the samples collected in 2005 and 2013. The overall prevalence of SNPs in *pfmdr-2* gene was 39.8% (74/186, 95%CI: 25.1–50.4). Of this, 4.2% (4/95), 76.9% (70/91) occurred in 2005 and 2013 samples respectively. In 2005 samples only one SNP, Y423F (4.2%, 4/95) was found while in 2013, Y423F (38.5%, 35/91) and I492V (38.5%, 35/91) SNPs in the *pfmdr-2* gene were found. There was a statistically significant difference in the prevalence of *pfmdr-2* SNPs in the samples collected in 2005 and 2013 ($p < 0.001$). The overall prevalence of *arps10* mutations was 2.7% (2/72, 95%CI: 0.3–4.2). Two mutations, V127M (4.5%: 1/22)

**Funding:** The review has received funding from a Grant, Number D43TW010132, supported by Ofce of the Director, National Institutes of Health (OD), National Institute of Dental & Craniofacial Research (NIDCR), National Institute of Neurological Disorders and Stroke (NINDS), National Heart, Lung, and Blood Institute (NHLBI), Fogarty International Center (FIC), and National Institute on Minority Health and Health Disparities (NIMHD). Additional funding was obtained from Malaria training Grant Number D43TW010526. The funders have had no role in the design of the study and in writing of the review protocol.

**Competing interests:** The authors have declared that no competing interests exist.

and D128H (4.5%: 1/22) in the *arps10* gene were each found in *P. falciparum* parasite samples collected in 2013. There was no statistically significant difference in the prevalence of *arps10* SNPs in the samples collected in 2005 and 2013 ($p = 0.238$). There were more *pfmdr-2* SNPs in *P. falciparum* parasites collected after introduction of Artemisinin combination therapies in malaria treatment. This is an indicator of the need for continuous surveillance to monitor emergence of molecular markers of artemisinin resistance and its potential drivers in malaria affected regions globally.

## Introduction

Uganda is among the 29 malaria endemic countries which contributed 95% of malaria cases globally in 2019 [1]. *P. falciparum* is responsible for over 99% of all malaria infections in Uganda [1, 2]. In sub-Saharan Africa, artemisinin combination therapies (ACTs) remain highly efficacious in malaria treatment despite the threat of parasite resistance to both artemisinin [1, 3] and other components of the combination in South east Asia [4]. This has contributed to the current success in malaria control efforts in the region [1]. In Uganda, the recommended first-line and second-line agents in treatment of uncomplicated malaria are Artemether-lumefantrine (AL) and Dihydroartemisin-Piperaquine (DHP-PPQ) respectively [5]. A recent systematic review reported a slightly better efficacy of DHP-PPQ compared to AL in the treatment of uncomplicated malaria among children under five years in Uganda [6].

Artemisinin resistance characterized by delayed parasite clearance is currently prevalent in Southeast Asia [7, 8]. From its early detection in Cambodia [9], artemisinin resistance has spread to other areas, including Thailand, Myanmar, China, Vietnam, Bangladesh, and Lao People's Democratic Republic [10, 11]. However, artemisinin resistance has not been detected outside China, Bangladesh and South east Asia for over a decade after it was first reported [1].

Parasite genetic background support independent emergence of K13 mutations associated with artemisinin resistance in different geographical areas in Southeast Asia [7, 8]. A study by Miotto *et al.*, [8] reported a strong association between non-synonymous mutations *fd*-D193Y (ferredoxin gene), *arps10*-V127M (apicoplast ribosomal protein S10 gene), *pfmdr2*-T484I (multidrug resistance protein 2 gene), *crt*-I356T (chloroquine resistance transporter gene) and *crt*-N326S (chloroquine resistance transporter gene) and development of artemisinin resistance in *P. falciparum* parasites [8]. Diverse K13 mutations whose role in artemisinin resistance is not yet known are prevalent in sub-Saharan Africa [12]. With the likely threat of emergence and spread of artemisinin resistance in sub-Saharan Africa, understanding the drivers of artemisinin resistance development is critical in guiding establishment of effective control and elimination strategies to combat resistance.

In highly malaria endemic regions like sub-Saharan Africa, development and spread of artemisinin resistance could result in increased malaria related mortality. A previous study [13], projected that emergence of wide spread artemisinin resistance globally could result in more than 100,000 malaria related deaths annually. Wide spread resistance to chloroquine followed by its subsequent withdrawal from malaria treatment in sub-Saharan Africa previously resulted in more than doubling of malaria related deaths [14]. Strategies to mitigate potential emergence of artemisinin resistance and thus prolong the effective life span of these agents in sub-Saharan Africa are urgently needed as they remain crucial for malaria control efforts. This is especially the case due to the current lack of an effective malaria vaccine.

*Plasmodium falciparum* parasite genetic background has been reported as an important driver of artemisinin resistance development in Southeast Asia [8, 15]. Recent studies [16, 17] have reported occurrence of K13 mutations 675V (Uganda and Rwanda) and 469Y (Uganda) that are associated with delayed artemisinin parasite clearance in Southeast Asia. Surveillance to monitor prevalence of markers of a genetic background which supports the rise of K13 mutations associated with delayed artemisinin parasite clearance across malaria endemic regions is thus crucial to help inform malaria control efforts. We therefore, intended in this study to assess and compare the prevalence of *fd*-D193Y, *aps10*-V127M, *pfmdr2*-T484I, *crt*-I356T, *crt*-N326S and K13 mutations in *P. falciparum* parasites before (2005) and after- (2013) introduction of artemisinin-based combination therapies for malaria treatment in Uganda [18].

## Materials and methods

### Ethics statement

The protocol was reviewed and approved by Makerere University School of Biomedical Ethics Review Committee (#SBS-513) and Uganda National Council of Science and Technology (#HS168ES). Written informed consent was obtained from the parent or guardian of the children prior to enrolment into the study.

### Study design, site and population

This was a cross sectional study. We used samples from two previous studies conducted by Infectious Disease Research Collaboration (IDRC); a randomized, single-blinded, longitudinal clinical trial designed to compare safety, tolerability and efficacy of three different combination antimalarial regimens for treatment of uncomplicated malaria done from November 2004 to June 2006 in Kampala [19]; and a cohort study, Program for Resistance, Immunology, Surveillance and Modeling of Malaria in Uganda (PRISM 2) conducted in three districts (Jinja, Kanungu and Tororo) from August 2011 to September 2013 [20]. In both studies malaria parasite infected blood samples were collected using ethylenediaminotetraacetic acid (EDTA) vacutainer tubes from children aged 1–10 years [19] and 6 months-10 years [20]. The two studies covered both low (Kampala, Kanungu) and high (Jinja, Tororo) malaria transmission settings in Uganda. The frozen whole blood samples were stored at the IDRC laboratory in Butabika National referral hospital and the research team accessed the samples in April 2019. All the samples were completely de-identified prior to being accessed by the research team. For our study, 100 randomly selected samples with ≥50ng/μl of DNA each collected in 2005 and 2013 by Dorsey *et al.*, [19] and Kamya *et al.*, [20] were included into the study. For the current study, we only used samples collected from Jinja and Kampala.

Microscopy, thick smears stained with 2% Giemsa for 30 minutes (detection of malaria parasite presence) and thin smear (malaria parasite species identification) were used for malaria diagnosis by the primary studies and were performed before sample storage [19, 20]. The thick and thin smears were prepared following a method previously described by WHO, 2004 [21]. In our current study, confirmation of the presence of *P. falciparum* parasites in the stored blood samples was done using malaria rapid diagnostic test (mRDT). The mRDT was done using *P. falciparum* histidine rich protein 2 (*PfHRP-2)* (Premier Medical Corporation Ltd, Gujarat, India) antigen based assay as described earlier by following the manufacturer's guidelines [22].

### DNA extraction and quantification

*P. falciparum* genomic DNA was extracted using QIAamp DNA mini kit (Qiagen, GmbH, Germany) following manufacturer's guidelines. The extracted genomic DNA concentration

was quantified using Nano drop spectrophotometry (Thermo Scientific, Wilmington, Delaware, USA). From the screening, 200 samples with ≥ 50 ng of *P. falciparum* parasite DNA were then included in the study.

## PCR amplification of P. falciparum K13 -propeller gene

The primers reported in a study by Ariey et al., [23] were double checked based on 3D7 genome using NCBI primer 3 software and synthesized by Eurofins scientific (S2 Table). Amplification of the K13 propeller domain of *P. falciparum* DNA was done following a method by Ariey *et al.*, [23]. Briefly, used in amplification of K13-propeller domain: 25µl reaction volume containing 12.5µl Kapa Hifi ready PCR ready mix (Roche), 1.0µl of each primer, 5.5µl of nuclease free water and 5µl of a mixture of both human and *P. falciparum* DNA (approximately 50ng/µL). Amplification was done in a Thermocycler under the following cycling conditions, 15 min at 95˚C, then 40 cycles of 30s at 94˚C, 90s at 54˚C, 90s at 72˚C and 10 min at 72˚C.

For the nested PCR, 2 µl of primary PCR products were amplified under the same conditions. The PCR products were detected using 2% agarose gel electrophoresis stained with ethidium bromide. The PCR products were shipped for Sanger dideoxy sequencing at ACGT Inc. (Wheeling IL, USA) commercial sequencing center.

## Amplification of P. falciparum crt, fd, arps10 and pfmdr-2 genes

Nested PCR approach was used to amplify portions of the *P. falciparum crt*, and *pfmdr-2* genes sandwiching the genetic background mutations *pfcrt* N326S, and *pfmdr-2* T484L. While for *fd*, and *arps10* genes the amplicons were generated using first round polymerase chain reaction (PCR). Briefly, used in amplification, 25µl reaction volume containing 12.5µl Kapa Hifi ready PCR ready mix (Roche), 1.0µl of each primer, 5.5µl of nuclease free water and 5µl of approximately 50ng/µL of *P. falciparum* DNA. Amplification was done using a Thermocycler following conditions as described in S1 Table. PCR products were detected using 2% agarose gel electrophoresis stained with ethidium bromide. Primers reported in a study by Miotto et al., [8] were double checked based on the 3D7 genome using NCBI primer 3 software and synthesized by Eurofins scientific (S2 Table). The PCR products were then shipped for Sanger dideoxy sequencing at ACGT Inc. (Wheeling IL, USA) commercial sequencing center.

## Sequence data analysis

Sequence data were base called using sequence analysis software Bioedit *ver* 5.2 and then blasted on to the NCBI sequence data base to confirm 3D7 K13 propeller, *arsp10*, *fd*, *pfmdr-2* and *pfcrt* gene sequence identity. The sequences were analyzed using Mutation Surveyor (Soft Genetics LLC., version 5.1, State College, PA, USA) in order to determine mixed alleles based on the presence of two chromatogram peaks at one nucleotide site, and to also reduce the presence of false positive and (or) false negative mutant sites [24, 25]. Fasta files were then aligned using UGENE v.39 (Unipro) and MEGA 5.10 software to reference allele sequences PF3D7_1343700 (K13), PF3D7_1460900.1 (*arps10*), PF3D7_1318100 (*fd*), PF3D7_1447900 (*pfmdr-2*) and PF3D7_070900 (*crt*) [24, 25] for detection of single nucleotide polymorphisms. Multiplicity of infection was assessed using the peak heights, where a mixed genotype was confirmed if the minor peak was higher than a third of the major peak.

We deposited sequences for mutations detected in the K13 gene (A578S, I587I, Y588N, E596V, S600C and E643K), two sequences for mutations in the *pfmdr-2* gene (Y423F, I492V) and two sequences for mutations in the *arps10* gene (V127M, D128H) in the NCBI data base.

The sanger traces were deposited in the GenBank and can be accessed using the accession numbers MZ668587-MZ668597 and MZ818701-MZ818770.

## Statistical analysis

Data analysis was done at 95% level of significance in STATA *ver* 14. The prevalence of mutations was determined using proportions. Correlation analysis was done using Fisher's exact test to assess the differences in prevalence of mutations in 2005 and 2013.

## Results

### Prevalence of K13 single nucleotide polymorphisms in the P. falciparum parasites

Of the 200 *P. falciparum* DNA samples screened for K13 mutations, 142 were successfully sequenced, 51 and 91 from 2005 and 2013 *P. falciparum* parasite samples respectively. Six (4.2%, 6/142) K13 SNPs were detected (Tables 1 and 2). Of these, five were non-synonymous and one was a synonymous K13 mutation, (I587I). We found four, one K13 non-synonymous mutations in the *P. falciparum* parasite samples collected in 2013 and 2005 respectively. K13 non-synonymous SNPs Y588N, E595V, S600C and E643K, each occurred in a single *P. falciparum* parasite sample. Two (2) K13 non-synonymous SNPs, A578S and E596V occurred in more than one *P. falciparum* parasite sample (Table 1). One K13 non-synonymous SNP Y588N occurred in a *P. falciparum* parasite sample collected in 2005. Five K13 non-synonymous SNPs A578S, E596V, S600C and E643K, were detected in *P. falciparum* parasite samples collected in 2013. There were no mixed infections identified among the samples.

Double non-synonymous K13 mutations, E596V and E643K were both found in the same *P. falciparum* parasite sample. In addition, a synonymous, I587I and a non-synonymous Y588N double K13 mutations were also both present in one *P. falciparum* parasite sample. One K13 synonymous SNP (I587I) was found in our study.

There was no statistically significant difference in the prevalence of mutations in the K13 ($p = 0.112$) and *arps10* ($p = 0.238$) genes for samples collected in 2005 and 2013. There was a

**Table 1. Single nucleotide polymorphisms identified in K13 gene in samples collected in Kampala and Jinja, Uganda in 2005 and 2013.**

| Sample identifier | Codon | Type | Reference amino acid | Mutant amino acid | Mutant locus | Reference allele | Mutant allele | n/N |
|---|---|---|---|---|---|---|---|---|
| K007 | 596[a],* | NS | E | V | 1787 | A | T | 3/142 |
| K007 | 643* | NS | E | K | 1927 | G | A | 1/142 |
| K020 | 600* | NS | S | C | 1799 | C | G | 1/142 |
| K065 | 578[b],* | NS | A | S | 1732 | G | T[c] | 2/142 |
| K120 | 587[e] | S | I | I | 1761 | T | C | 1/142 |
| K120 | 588[d,e] | NS | Y | N | 1762 | T | A | 1/142 |

N: Number of samples sequenced at locus

n: Number of samples containing mutant allele

S: Synonymous SNP

NS: Non-synonymous SNP

[a] Occurred in three *P. falciparum* parasite samples (K007, K050, K082)

[b] Occurred in two *P. falciparum* parasite samples (K065, K070)

[c] SNP has previously been identified (MalariaGen)

[d] Different SNP in the same codon position has been previously reported (MalariaGen)

*SNP occurred in samples collected in 2013

[e] SNP occurred in samples collected in 2005

**Table 2. Prevalence of single nucleotide polymorphisms in *K13, Pfmdr-2 and Pfarps 10* genes in Plasmodium falciparum samples collected in Kampala and Jinja, Uganda in 2005 and 2013.**

| Characteristic | Had mutation n | No mutation n | Description | Prevalence of mutation %(n/N) | Fisher's Exact P-value | 95% CI |
|---|---|---|---|---|---|---|
| K13 SNPs (overall) | 6 | 136 | Overall | 4.2 (6/142) | 0.112 | 1.4–7.0 |
| | 2 | 49 | 2005 | 3.9 (2/51) | | 0.4–8.2 |
| | 4 | 87 | 2013 | 4.4 (4/91) | | 2.5–12.6 |
| K13 SNPs (2005) | 1 | 50 | Y588N | 1.9 (1/51) | | 0.1–7.2 |
| | 1 | 50 | I587I | 1.9 (1/51) | | 0.1–7.2 |
| K13 SNPs (2013) | 3 | 88 | E596V | 3.3 (3/91) | | 1.1–9.8 |
| | 1 | 90 | E643K | 1.1 (1/91) | | 0.2–7.6 |
| | 1 | 90 | S600C | 1.1(1/91) | | 0.2–7.6 |
| | 2 | 89 | A578S | 2.2(2/91) | | 0.5–8.5 |
| *pfmdr-2* SNPs (overall) | 74 | 112 | Overall | 39.8 (74/186) | <0.001 | 25.1–50.4 |
| | 4 | 91 | 2005 | 4.2 (4/95) | | 1.6–10.8 |
| | 70 | 21 | 2013 | 76.9 (70/91) | | 34.0–54.4 |
| *pfmdr-2* SNPs (2005) | 0 | 95 | T484I | 0(0) | | - |
| | 0 | 95 | I492V | 0(0) | | - |
| | 4 | 91 | Y423F | 4.2 (4/95) | | 1.6–10.8 |
| *pfmdr-2* SNPs (2013) | 0 | 91 | T484I | 0(0) | | - |
| | 35 | 56 | I492V | 38.5 (35/91) | | 12.0–39.0 |
| | 35 | 56 | Y423F | 38.5 (35/91) | | 12.0–39.0 |
| *pfarps10* SNPs (overall) | 2 | 72 | Overall | 2.7 (2/74) | 0.238 | 0.3–4.2 |
| | 0 | 52 | 2005 | 0 (0) | | - |
| | 2 | 22 | 2013 | 8.3 (2/24) | | 1.6–19.5 |
| *pfarps10* SNPs (2005) | 0 | 52 | V127M | 0 (0) | | - |
| | 0 | 52 | D128H | 0 (0) | | - |
| *pfarps10* SNPs (2013) | 1 | 21 | V127M | 4.5 (1/22) | | 1.1–10.5 |
| | 1 | 21 | D128H | 4.5 (1/22) | | 1.1–10.5 |

SNPs: Single Nucleotide Polymorphisms; CI: Confidence Interval, n: number of samples with a given characteristic, N = total number of samples analysed, %: Percentage

statistically significant difference in the prevalence of mutations in the *pfmdr-2* gene, p<0.001 among samples collected in 2005 and 2013 (Table 2).

## Prevalence of genetic background mutations, pfcrt N326S, fd D193Y, arps10 V127M and pfmdr-2 T484L in P. falciparum parasites in Uganda

Sequencing of *pfmdr-2, fd, pfcrt* and *arps10* was successful in 186, 141, 128 and 74 *P. falciparum* DNA samples respectively. Polymorphisms in the *fd, pfmdr-2* and *pfcrt* genes that are markers of a genetic background where K13 mutations associated with slow artemisinin parasite clearance are likely to arise were not found in our study. The prevalence of K13 mutations was 4.2% (6/142; 95%CI: 1.4–7.0). We found one non-synonymous, Y588N (1.9%, 1/51; 95%CI: 0.1–7.2) and one synonymous K13 mutation, I587I (1.9%, 1/51; 95%CI: 0.1–7.2) in samples collected in 2005. While in *P. falciparum* from samples collected in 2013, we found three non-synonymous K13 mutations, E596V (3.3%, 3/91; 95CI: 1.1–9.8), E643K (1.1%, 1/91; 95%CI: 1.1–9.8) and A578S (2.2%, 2/91; 95%CI: 0.5–8.5). The overall prevalence of *pfmdr-2* mutations was 39.8% (74/186; 95%CI: 25.1–50.4). In the *pfmdr-2* gene, we found two non-synonymous SNPs, Y423F and I492V in *P. falciparum* parasite samples. The SNP Y423F occurred in 4 (4.2%, 4/95; 95%CI: 1.6–10.8), 35(38.5%, 35/91; 95%CI: 12.0–39.0) *P. falciparum* parasite samples collected

in 2005 and 2013 respectively. The SNP, I492V occurred in 35 (38.5%, 35/91; 95%CI: 12.0–39.0) *P. falciparum* samples all collected in 2013. In the *arps10* gene, we found two (2.7%, 2/74; 95%CI: 0.3–4.2) non-synonymous SNPs, V127M (4.5%, 1/22; 95%CI: 1.1–10.5) and D128H (4.5%, 1/22; 95%CI: 1.1–10.5) each in one *P. falciparum* parasite sample collected in 2013 (Table 2). The two *P. falciparum* parasite samples with *arps10* gene mutations did not have K13 gene mutation. Two *P. falciparum* parasite samples collected in 2013 had mutations in both *pfmdr-2* (Y423F) and K13 (A578S) genes.

## Discussion

*Plasmodium falciparum* artemisinin resistance is a heritable trait with a genetic basis [7]. Genome modification studies have shown that the impact of various K13 mutations on *P. falciparum* artemisinin clearance and survival rates of ring stage parasites is dependent on the genetic background [26]. The risk of emergence of K13 mutations associated with delayed artemisinin parasite clearance is thus driven by specific parasite genetic background [8, 27]. In our current study polymorphisms in the *fd*, *pfmdr-2* and *pfcrt* genes which support the rise of K13 mutations associated with delayed artemisinin clearance of *P. falciparum* parasites reported in Southeast Asia [8] were not found. However, SNPs in the *pfmdr-2*, *pfcrt*, and *arps10* genes not reported in a study by Miotto *et al.*, [8] and other previous studies in Africa [28] were detected in our current study. There is need to validate the role the identified background mutations play in artemisinin resistance development among African *P. falciparum* parasites.

Artemisinin resistance has been shown to arise independently in different settings in Southeast Asia [8, 11], an indicator of the role unique sets of conditions in different malaria endemic regions play in driving resistance development. In our study, we detected background mutations in the *pfmdr-2*, I492V (38.5%, 35/91; 95%CI: 12.0–39.0) and Y423F (2005: 4.2%, 4/95; 95%CI: 1.6–10.8; 2013: 38.5%, 35/91; 95%CI: 12.0–39.0) and *arps10* (D128H; 4.5%, 1/22; 95% CI: 1.1–10.5)*)* genes whose role in artemisinin resistance is not known. These mutations were found in *P. falciparum* parasites collected in 2013 after introduction of artemisinin combination therapies for malaria treatment in Uganda [18]. There is however a need for more surveillance studies to establish *P. falciparum* parasite genetic background in the country. This is especially critical as previous studies done in Southeast Asia have demonstrated an association between parasite genetic background and the independent emergence of artemisinin resistance [8, 11]. A recent study in South Sudan [29] which analyzed samples collected in 2015–2017 after introduction of artemisinin agents in malaria treatment detected genetic background mutation in *Pfcrt* N326S gene which was previously reported in Southeast Asia and associated with artemisinin resistance. While we did not find in our samples PfcrtN326S SNP, we detected D128H after introduction of artemisinin agents in malaria treatment. Additionally, we detected a background mutation, V127M in the *arps10* gene that has been shown to support development of K13 mutations associated with artemisinin resistance in Southeast Asia [8]. However, it still remains unknown what role other background mutations (*arps10* D128H, *pfmdr-2*-Y423F, *pfmdr-2*-I492V) found in our study play in driving independent development and spread of artemisinin resistance among *P. falciparum* parasites in sub-Saharan Africa.

We detected different K13 SNPs in *P. falciparum* samples collected in 2005 compared to those collected in 2013 after introduction of artemisinin combination therapies in malaria treatment in Uganda. Although not statistically significant, our study found a general trend towards increase in the prevalence of K13 SNPs after introduction of artemisinin agents in malaria treatment in Uganda. A previous study by Conrad *et al.*, [30] in Uganda showed that introduction of ACTs in malaria treatment did not increase the diversity of K13 SNPs among

*P. falciparum* parasites despite an overall increase in the prevalence. However, the role of these mutations in artemisinin resistance remains unknown. Recent studies in Rwanda [17, 31] and Uganda [16, 32, 33] have reported presence of K13 mutations 675V and 561H (Rwanda), 675V and 469Y (Uganda) associated with delayed artemisinin parasite clearance in Southeast Asia. A recent study by Balikagala *et al.*, [33] demonstrated delayed artemisinin clearance among parasites carrying 675V and 469Y mutations in Uganda. The discovery of these mutations in Ugandan *P. falciparum* parasites potentially threatens the efficacy of artemisinin combination therapies, a cornerstone in malaria treatment [1]. There is thus an urgent need to investigate the mediators driving development and spread of these mutations in the country.

Our study had some limitations, we analyzed few samples however inclusion of *P. falciparum* parasites collected in both high and low malaria transmission settings in the country helped improve representativeness. In addition, analysis of *P. falciparum* parasites collected prior to an after introduction of artemisinin combination therapies in malaria treatment in Uganda helped provide information on potential drivers of the prevalence of K13 and genetic background mutations in the country. Our study could not confirm existence of multiplicity of infection in the samples analyzed.

## Conclusion

A mutation, V127M in the *arps10* gene previously reported in South east Asia and associated with delayed artemisinin parasite clearance was detected in *P. falciparum* parasites in Uganda. The proportion of *pfmdr-2* gene mutations were generally higher in *P. falciparum* parasites collected after introduction of artemisinin combination therapies in malaria treatment in Uganda. There were neither K13 SNPs previously implicated in artemisinin resistance in Africa and SEA, nor mutations in the associated background genes identified in this study with exception of one SNP (V127M) in the *arps10* gene. There is need to conduct regular surveillance to monitor potential emergence of molecular markers of artemisinin resistance and their drivers among *P. falciparum* parasites in malaria affected regions globally.

## Supporting information

**S1 Table. PCR cycling conditions for amplification of *P. falciparum* DNA fragments sandwiching *pfcrt* N326S, *fd* D193Y, *arps10* V127M and *mdr2* T484L genetic background mutations.**
(DOC)

**S2 Table. Primer sets used during amplification of *Plasmodium falciparum* DNA.**
(DOC)

## Acknowledgments

We acknowledge the guidance of the laboratory team at the Infectious Disease Research Collaboration (IDRC)-Makerere University for their guidance.

## Author Contributions

**Conceptualization:** Moses Ocan, Moses R. Kamya, Sam L. Nsobya.

**Data curation:** Moses Ocan.

**Formal analysis:** Moses Ocan, Fred Katabazi Ashaba, Savannah Mwesigwa, Kigozi Edgar, Sam L. Nsobya.

**Investigation:** Fred Katabazi Ashaba.

**Methodology:** Kigozi Edgar, Moses R. Kamya.

**Writing – original draft:** Moses Ocan.

**Writing – review & editing:** Moses Ocan, Fred Katabazi Ashaba, Savannah Mwesigwa, Kigozi Edgar, Moses R. Kamya, Sam L. Nsobya.

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
