## [Decision Letter · Decision Letter 0]

13 Dec 2021

PONE-D-21-33254Prevalence of arps10, fd, mdr-2 and pfkelch13 gene mutations in Plasmodium falciparum parasite population in UgandaPLOS ONE

Dear Dr. Ocan,

Thank you for submitting your manuscript to PLoS ONE. After careful consideration, we felt that your manuscript requires revision, following which it can possibly be reconsidered. As quoted by a different reviewer, major concerns were still related to study design and data presentation. According to the reviewers, the methods were not described in enough details to allow suitably skilled investigators to fully replicate and evaluate the study. In addition, a significant number of issues should be clarified and/or adjust otherwise the MS’s results may be compromised. For your guidance, a copy of the reviewers' comments was included below

We look forward to receiving your revised manuscript.

Kind regards,

Luzia Helena Carvalho, Ph.D.

Academic Editor

PLOS ONE

2. Please provide additional details regarding participant consent. In the Methods section, please ensure that you have specified (1) whether consent was informed and (2) what type you obtained (for instance, written or verbal). If your study included minors, state whether you obtained consent from parents or guardians. If the need for consent was waived by the ethics committee, please include this information.

Reviewers' comments:

Reviewer's Responses to Questions

**Comments to the Author**

1. Is the manuscript technically sound, and do the data support the conclusions?

Reviewer #1: Partly

Reviewer #2: Partly

2. Has the statistical analysis been performed appropriately and rigorously? 

Reviewer #1: No

Reviewer #2: No

3. Have the authors made all data underlying the findings in their manuscript fully available?

Reviewer #1: Yes

Reviewer #2: Yes

4. Is the manuscript presented in an intelligible fashion and written in standard English?

Reviewer #1: Yes

Reviewer #2: Yes

5. Review Comments to the Author

Reviewer #1: Comments:

The MS “Prevalence of arps10, fd, mdr-2, pfcrt and pfkelch13 gene mutations in Plasmodium falciparum parasite population in Uganda” by Moses Ocan et al., presents useful molecular surveillance information on the polymorphisms in the genes responsible for driving development of resistance to artemisinins.

Major comments:

1) Abstract/Results

It is difficult to follow the results description in the Abstract and Results as there is no clear demarcation between the SNP prevalence before 2005 and after 2015, because the authors report the prevalences in the pooled sample set. In my view, the analysis (in the Results and the Abstract) should be presented for each cohort separately (as a Table) and basic statistical analysis (presenting P values and confidence intervals) should be done to claim any significant difference in the point prevalences of SNPs before and after introduction of the ACTs. In the current version it is not clear if the following statement in lines 52-53 can be made “This is an indicator of the role Artemisinin combination therapies play in altering P. falciparum parasite genotype and potentially driving resistance development”. I suggest to change it emphasizing the importance of molecular surveillance in order to detect emerging resistance.

2) Introduction still needs more information on ACTs used in these areas, their efficacies and any evidence of resistance developing. Recommend to add references such as Assefa, D.G., Zeleke, E.D., Bekele, D. et al. Efficacy and safety of dihydroartemisinin–piperaquine versus artemether–lumefantrine for treatment of uncomplicated Plasmodium falciparum malaria in Ugandan children: a systematic review and meta-analysis of randomized control trials. Malar J 20, 174 (2021).

3) Information on molecular markers of artemisinin resistance in Africa needs to be added, for example, found in recent refs: Tumwebaze PK, Katairo T, Okitwi M, Byaruhanga O, Orena S, Asua V, Duvalsaint M, Legac J, Chelebieva S, Ceja FG, Rasmussen SA, Conrad MD, Nsobya SL, Aydemir O, Bailey JA, Bayles BR, Rosenthal PJ, Cooper RA. Drug susceptibility of Plasmodium falciparum in eastern Uganda: a longitudinal phenotypic and genotypic study. Lancet Microbe. 2021 Sep;2(9):e441-e449.

4) There is also a similar study (Conrad MD, Nsobya SL, Rosenthal PJ. The Diversity of the Plasmodium falciparum K13 Propeller Domain Did Not Increase after Implementation of Artemisinin-Based Combination Therapy in Uganda. Antimicrob Agents Chemother. 2019;63(10):e01234-19.) conducted in the same areas, where the authors found “No differences in diversity following implementation of ACT use were found at any of the seven sites, nor was there evidence of selective pressures acting on the locus. Our results suggest that selection by ACTs is not impacting on K13PD diversity in Uganda”. This ref and information should be discussed in Introduction and Discussion.

5) Materials and Methods. Study collection times need to be clarified.

Lines 124-125: Suggest specify enrolment time as in ref 16: “… with enrolment conducted from November 2004 to April 2005 in Kampala (16).

Ref 17 has incorrect publication year 2020 instead of 2015 in

Kamya MR, Arinaitwe E, Wanzira H, Katureebe A, Barusya C, Kigozi SP, et al. Malaria Transmission, Infection, and Disease at Three Sites with Varied Transmission Intensity in Uganda: Implications for Malaria Control. Am J Trop Med Hyg. 2020;92(5):903-12.

It should be “Malaria transmission, infection, and disease at three sites with varied transmission intensity in Uganda: implications for malaria control.

Kamya MR, Arinaitwe E, Wanzira H, Katureebe A, Barusya C, Kigozi SP, Kilama M, Tatem AJ, Rosenthal PJ, Drakeley C, Lindsay SW, Staedke SG, Smith DL, Greenhouse B, Dorsey G. Am J Trop Med Hyg. 2015 May;92(5):903-12”.

This study collection time is specified in this reference as “between August 2011 to September 2013” not “to 2018” as stated in the MS. This requires further clarification.

As only a subset from these studies being used in the present study the detailed description of how many samples from each site were included in either cohort would be beneficial.

6) Line 134 “For our study, 100 P. falciparum infected blood samples each collected in 2005 and 2015 by Dorsey et al., (16) and Kamya et al., (17), respectively were randomly selected for inclusion. In addition, only samples with ≥50ng/µl of DNA were included in the study.”

The statement needs to be rephrased as samples were not exactly collected in 2005 or 2015. This needs to be addressed throughout the entire text.

How many samples from each site were included and how randomization was achieved?

Also suggest to change to “100 randomly selected samples with ≥50ng/µl of DNA were included into the study” .

6). Line 155: It is not clear if the primers for K13 amplification were designed by the authors or just synthesized based on previously published ones by Ariey et al. 2014? Did authors designed the primers for other genes SNPs?

7) Supplement A includes the tables also included in S1 and S2 supplementary files, which is redundant.

8) Lines 193 and 218” Authors refer to carrying “amplicon sequencing”, perhaps, meaning “PCR amplicon”, which is a little bit confusing since this term is typically used for deep amplicon sequencing, where the libraries are generated from a PCR product and sequenced to identify rare mutations. In this study it appears that authors have carried out Sanger sequencing of PCR products. If that is the case, I suggest, that the authors remove “amplicon” and refer to their sequencing methodology as “Sanger sequencing of PCR products”. It also follows, that it is not possible to determine the multiplicity of infections by analysing the Sanger sequencing chromatograms peaks (e.g., of K13 genes). It would be possible to determine the dominant alleles, but this method lacks sensitivity and unable to determine minor alleles (as outlined in Zhong, D., Koepfli, C., Cui, L. et al. Molecular approaches to determine the multiplicity of Plasmodium infections. Malar J 17, 172 (2018). Analysis of highly polymorphic markers (genes or microsattelites) is used to estimate MOI.

9) Lines 302… The statement in conclusions is correct but without statistical analysis can be misleading: “The proportions of pfkelch13, arsp10, and mdr-2 gene mutations were higher in P. falciparum parasites collected a decade after introduction of artemisinin combination therapies in malaria treatment in Uganda.”

Minor comments:

1) For consistency, the genes’ abbreviations need to be the same throughout the text, for an example mdr-2 or pfmdr2. There are typos in arps10 gene name in lines 85 and 109.

2) Likewise, aminoacid abbreviations used in some cases are 3-letter ones whereas in others are 1-letter abbreviation -needs to be consistent throughout the text.

2). It is more appropriate to use single nucleotide polymorphism (SNP) term rather than mutations, for an example when referring to K13 polymorphisms.

3). Ref 13 and 27 are the same.

4) Lines 96 “Wide spread chloroquine resistance followed by its subsequent withdrawal…” suggest to change to “Wide spread of resistance to chloroquine followed by its subsequent withdrawal…”

Line 148: add GmbH and the country information to Qiagen company reference in the following sentence: “P. falciparum genomic DNA was extracted using QIAamp DNA mini kit (Qiagen, GmbH, Germany).

Reviewer #2: Reviewer’s comments:

The authors describe the prevalence of Pfkelch 13 gene mutations and other background genetic mutations thought to associate with artemisinin resistance. The author utilised available samples at their disposal to answer the research question. While I appreciate the efforts of the authors, I shall request that the following corrections be made before acceptance:

#1- The author should state clearly if the mutations found in the Pfkelch 13 gene were among associated to artemisinin resistance. This should be clear as from abstract through discussion.

#2- The author should so kindly introduce some statistical analysis like fishers’ exact test or chi square to compare the proportion of mutations before and after introduction of ACT in Uganda in his samples. This is important!!!

Line 42#- the sentence should be corrected.

Line 81-82# I went through the reference provided by the author and it appears that it didn’t support the statement. The author should find an appropriate reference to that statement.

Line 217# . Could the author please make a clear table showing the mutations that occurred in 2005 and that which occurred in 2015. The table is not interesting in its current form and does not represent the line of thoughts in the paper.

Line 254 -225#- The author seems to contradict his basis for accessing the background mutations. In line 81-86, the author mentioned that the background mutations were associated with artemisinin resistance in different geographical areas in Southeast Asia, but here he presents that their role in artemisinin resistance is not known. I suggest that the authors should think around this and rephrase accordingly.

Line 298-300# - The author should support the statement with statistics.

Line 295-300#- The author should fit in one or two implications of his findings in the conclusion to make a bit stronger.

6. PLOS authors have the option to publish the peer review history of their article (what does this mean?). If published, this will include your full peer review and any attached files.

Reviewer #1: No

Reviewer #2: **Yes: **Ikegbunam Moses (PhD)

---

## [Author Response · Author response to Decision Letter 0]

5 Jan 2022

RESPONSE TO THE REVIEWER’S COMMENTS

Reviewer #1: Comments:

The MS “Prevalence of arps10, fd, mdr-2, pfcrt and pfkelch13 gene mutations in Plasmodium falciparum parasite population in Uganda” by Moses Ocan et al., presents useful molecular surveillance information on the polymorphisms in the genes responsible for driving development of resistance to artemisinins.

Response: Thanks for the comment

Reviewer #1: Comments: Major comments:

1) Abstract/Results

It is difficult to follow the results description in the Abstract and Results as there is no clear demarcation between the SNP prevalence before 2005 and after 2015, because the authors report the prevalences in the pooled sample set. In my view, the analysis (in the Results and the Abstract) should be presented for each cohort separately (as a Table) and basic statistical analysis (presenting P values and confidence intervals) should be done to claim any significant difference in the point prevalences of SNPs before and after introduction of the ACTs. In the current version it is not clear if the following statement in lines 52-53 can be made “This is an indicator of the role Artemisinin combination therapies play in altering P. falciparum parasite genotype and potentially driving resistance development”. I suggest to change it emphasizing the importance of molecular surveillance in order to detect emerging resistance.

Response: Re-analysis of the data has been done to reflect prevalence of mutations in the different years. For K13 SNPs overall prevalence is 4.2% (6/142) while in 2005 (1.4%, 2/142) and 2013 (2.8%, 4/142). For the pfmdr-2 mutations, overall prevalence is 39.8% (74/186) while in 2005 (2.2%, 4/186) and in 2013 (37.6%, 70/186). For arps10 mutations, overall prevalence is 2.7% (2/74) while in 2013 (2.7%, 2/74) and this mutation was not detected among samples collected in 2005. There was no significant relationship in the prevalence of mutations in the K13 (p=0.087) and arps10 (p=0.146) genes for samples collected in 2005 and 2013. There was a significant relationship in the prevalence of mutations in the pfmdr-2 gene, p<0.001 among samples collected in 2005 and 2013. 

Detailed results on this analysis is presented in Table 4 in the revised manuscript. 

This revision has been effected in the revised manuscript. 

Reviewer #1: Comments: 2) Introduction still needs more information on ACTs used in these areas, their efficacies and any evidence of resistance developing. Recommend to add references such as Assefa, D.G., Zeleke, E.D., Bekele, D. et al. Efficacy and safety of dihydroartemisinin–piperaquine versus artemether–lumefantrine for treatment of uncomplicated Plasmodium falciparum malaria in Ugandan children: a systematic review and meta-analysis of randomized control trials. Malar J 20, 174 (2021).

Response: Thanks for the comment, the requested information is mentioned in the first paragraph of the manuscript. However, we have added further information on the different ACTs used in Uganda. In Uganda, the recommended first-line and second-line agents in treatment of uncomplicated malaria are Artemether-lumefantrine (AL) and Dihydroartemisin-Piperaquine (DHP-PPQ) respectively (MoH, 2011). A recent systematic review reported a slightly better efficacy of DHP-PPQ compared to AL in the treatment of uncomplicated malaria among children under five years in Uganda (Assefa et al., 2021). 

This has been incorporated in the revised manuscript. 

Reviewer #1: Comments: 3) Information on molecular markers of artemisinin resistance in Africa needs to be added, for example, found in recent refs: Tumwebaze PK, Katairo T, Okitwi M, Byaruhanga O, Orena S, Asua V, Duvalsaint M, Legac J, Chelebieva S, Ceja FG, Rasmussen SA, Conrad MD, Nsobya SL, Aydemir O, Bailey JA, Bayles BR, Rosenthal PJ, Cooper RA. Drug susceptibility of Plasmodium falciparum in eastern Uganda: a longitudinal phenotypic and genotypic study. Lancet Microbe. 2021 Sep;2(9):e441-e449.

4) There is also a similar study (Conrad MD, Nsobya SL, Rosenthal PJ. The Diversity of the Plasmodium falciparum K13 Propeller Domain Did Not Increase after Implementation of Artemisinin-Based Combination Therapy in Uganda. Antimicrob Agents Chemother. 2019;63(10):e01234-19.) conducted in the same areas, where the authors found “No differences in diversity following implementation of ACT use were found at any of the seven sites, nor was there evidence of selective pressures acting on the locus. Our results suggest that selection by ACTs is not impacting on K13PD diversity in Uganda”. This ref and information should be discussed in Introduction and Discussion.

Response: According to a number of studies, K13 mutations in Africa are very diverse and its probable that each study is likely to find slightly different K13 mutations. A statement on the diversity of K13 mutations in sub-Saharan Africa is provided in the earlier version of the manuscript and we think that this is sufficient information based on the current evidence around K13 mutations in Africa. Additionally, a paper by Conrad et al., 2019, reports no increase in the diversity of K13 mutations however, like our current study the article finds a general increase in the prevalence of K13 mutations in the period after introduction of ACTs in malaria treatment in Uganda. A statement on the relationship between introduction of ACTs and the diversity of K13 mutations has been incorporated in the revised manuscript. 

This information has been incorporated in the revised manuscript 

Reviewer #1: Comments 5) Materials and Methods. Study collection times need to be clarified.

Lines 124-125: Suggest specify enrolment time as in ref 16: “… with enrolment conducted from November 2004 to April 2005 in Kampala (16).

Response: Yes, sorry for the lack of clarity on this. We picked samples for our current study from those which were collected in 2005. This statement has been incorporated in the revised manuscript. 

Reviewer #1: Comments: Ref 17 has incorrect publication year 2020 instead of 2015 in

Kamya MR, Arinaitwe E, Wanzira H, Katureebe A, Barusya C, Kigozi SP, et al. Malaria Transmission, Infection, and Disease at Three Sites with Varied Transmission Intensity in Uganda: Implications for Malaria Control. Am J Trop Med Hyg. 2020;92(5):903-12.

It should be “Malaria transmission, infection, and disease at three sites with varied transmission intensity in Uganda: implications for malaria control.

Response: Thanks for the comment. This has been corrected in the revised manuscript. 

Reviewer #1: Comments: Kamya MR, Arinaitwe E, Wanzira H, Katureebe A, Barusya C, Kigozi SP, Kilama M, Tatem AJ, Rosenthal PJ, Drakeley C, Lindsay SW, Staedke SG, Smith DL, Greenhouse B, Dorsey G. Am J Trop Med Hyg. 2015 May;92(5):903-12”.

This study collection time is specified in this reference as “between August 2011 to September 2013” not “to 2018” as stated in the MS. This requires further clarification.

Response: Thanks for the comment. We are sorry for the lack of clarity on this. The dates for sample collections have been corrected. For our study we analyzed samples collected in 2013. This has been clarified in the revised manuscript. 

Reviewer #1: Comments: As only a subset from these studies being used in the present study the detailed description of how many samples from each site were included in either cohort would be beneficial.

Response: In our current study only samples from Jinja and Kampala were used. Of which 100 samples each from Jinja and Kampala were picked. 

This has been provided in the revised manuscript. 

Reviewer #1: Comments: 6) Line 134 “For our study, 100 P. falciparum infected blood samples each collected in 2005 and 2015 by Dorsey et al., (16) and Kamya et al., (17), respectively were randomly selected for inclusion. In addition, only samples with ≥50ng/µl of DNA were included in the study.”

The statement needs to be rephrased as samples were not exactly collected in 2005 or 2015. This needs to be addressed throughout the entire text.

Response: Thanks for the comment, a study by Dorsey et al., (16), a single-blind randomized clinical trial was conducted between November 2004 and June 2006. In our current study, were accessed 100 blood samples collected in 2005. For a study by Kamya et al., (17), the study was conducted from August 5, 2011 to September 30, 2013. We accessed 100 blood samples collected in 2013. This has been corrected in the revised manuscript. 

Reviewer #1: Comments: How many samples from each site were included and how randomization was achieved?

Also suggest to change to “100 randomly selected samples with ≥50ng/µl of DNA were included into the study” .

Response: A study by Kamya et al., (17) was part of a larger PRISM 2 study which was conducted in three districts of Jinja, Kanungu and Tororo while the study by Dorsey et al., (16) was conducted only in Kampala. For our current study, only 100 samples each from Jinja and Kampala were used. This has been incorporated in the revised manuscript. 

Reviewer #1: Comments: 6). Line 155: It is not clear if the primers for K13 amplification were designed by the authors or just synthesized based on previously published ones by Ariey et al. 2014? Did authors designed the primers for other genes SNPs?

Response: Thanks for the comment. For K13 gene, we synthesized the primers based on previously published primers in a study by Ariey et al., 2014. For pfmdr-2, arps10, fd and pfcrt genes, the primers published in a study by Miotto et al., were double checked using NCBI primer 3 software. The primers reported in a study by Miotto et al., were then synthesized. All the primers were synthesized by Eurofins scientific. This has been corrected in the revised manuscript. 

Reviewer #1: Comments :7) Supplement A includes the tables also included in S1 and S2 supplementary files, which is redundant.

Response: Thanks for the observation, ‘Supplement A’ has been removed/deleted from the manuscript files in the revised submission. 

Reviewer #1: Comments: 8) Lines 193 and 218” Authors refer to carrying “amplicon sequencing”, perhaps, meaning “PCR amplicon”, which is a little bit confusing since this term is typically used for deep amplicon sequencing, where the libraries are generated from a PCR product and sequenced to identify rare mutations. In this study it appears that authors have carried out Sanger sequencing of PCR products. If that is the case, I suggest, that the authors remove “amplicon” and refer to their sequencing methodology as “Sanger sequencing of PCR products”.

Response: Thanks for the comments, this has been corrected in the revised manuscript. The statement now reads, ‘Sanger sequencing of PCR products’. 

Reviewer #1: Comments: It also follows, that it is not possible to determine the multiplicity of infections by analysing the Sanger sequencing chromatograms peaks (e.g., of K13 genes). It would be possible to determine the dominant alleles, but this method lacks sensitivity and unable to determine minor alleles (as outlined in Zhong, D., Koepfli, C., Cui, L. et al. Molecular approaches to determine the multiplicity of Plasmodium infections. Malar J 17, 172 (2018). Analysis of highly polymorphic markers (genes or microsattelites) is used to estimate MOI.

9) Lines 302… The statement in conclusions is correct but without statistical analysis can be misleading: “The proportions of pfkelch13, arsp10, and mdr-2 gene mutations were higher in P. falciparum parasites collected a decade after introduction of artemisinin combination therapies in malaria treatment in Uganda.”

Response: The limitation of analyzing Sanger sequencing chromatogram peaks in determining multiplicity of infections has been included in the revised manuscript. We have performed statistical analysis of the data to support the statement in the conclusion. 

Minor comments:

Reviewer #1: Comments: 1) For consistency, the genes’ abbreviations need to be the same throughout the text, for an example mdr-2 or pfmdr2. There are typos in arps10 gene name in lines 85 and 109.

Response: Thanks for the comment, this has been adjusted in the revised manuscript. 

Reviewer #1: Comments: 2) Likewise, aminoacid abbreviations used in some cases are 3-letter ones whereas in others are 1-letter abbreviation -needs to be consistent throughout the text.

Response: This has been corrected in the revised manuscript. 

Reviewer #1: Comments: 2). It is more appropriate to use single nucleotide polymorphism (SNP) term rather than mutations, for an example when referring to K13 polymorphisms.

Response: This has been adjusted in the revised manuscript 

Reviewer #1: Comments: 3). Ref 13 and 27 are the same.

Response: Thanks for the comment, this has been corrected in the revised manuscript. 

Reviewer #1: Comments: 4) Lines 96 “Wide spread chloroquine resistance followed by its subsequent withdrawal…” suggest to change to “Wide spread of resistance to chloroquine followed by its subsequent withdrawal…”

Response: Thanks, this has been changed to “Wide spread of resistance to chloroquine followed by its subsequent withdrawal…”, in the revised manuscript. 

Reviewer #1: Comments: Line 148: add GmbH and the country information to Qiagen company reference in the following sentence: “P. falciparum genomic DNA was extracted using QIAamp DNA mini kit (Qiagen, GmbH, Germany).

Response: Thanks for the comment, this has been revised to “P. falciparum genomic DNA was extracted using QIAamp DNA mini kit (Qiagen, GmbH, Germany) in the corrected manuscript. 

Reviewer #2: Reviewer’s comments:

The authors describe the prevalence of Pfkelch 13 gene mutations and other background genetic mutations thought to associate with artemisinin resistance. The author utilised available samples at their disposal to answer the research question. While I appreciate the efforts of the authors, I shall request that the following corrections be made before acceptance:

#1- The author should state clearly if the mutations found in the Pfkelch 13 gene were among associated to artemisinin resistance. This should be clear as from abstract through discussion.

Response: Thanks for the comments, in our manuscript this information is clearly stated in the discussion section. The K13 mutations found in our study have not among those that have been validated to be associated with artemisinin resistance. 

Reviewer #2: Reviewer’s comments: #2- The author should so kindly introduce some statistical analysis like fishers’ exact test or chi square to compare the proportion of mutations before and after introduction of ACT in Uganda in his samples. This is important!!!

Response: Thanks, we have performed statistical analysis and have incorporated the findings in the results section of the manuscript. 

Reviewer #2: Reviewer’s comments Line 42#- the sentence should be corrected.

Response: Thanks, we have corrected the sentence. 

Reviewer #2: Reviewer’s comments: Line 81-82# I went through the reference provided by the author and it appears that it didn’t support the statement. The author should find an appropriate reference to that statement.

Response: Thanks for the comment, we have provided an alternative reference in the revised manuscript. 

Reviewer #2: Reviewer’s comments: Line 217#. Could the author please make a clear table showing the mutations that occurred in 2005 and that which occurred in 2015. The table is not interesting in its current form and does not represent the line of thoughts in the paper.

Response: Thanks for the comment, the mutations that occurred in 2005 and 2015 have been clearly indicated in the table in the revised manuscript. 

Reviewer #2: Reviewer’s comments: Line 254 -225#- The author seems to contradict his basis for accessing the background mutations. In line 81-86, the author mentioned that the background mutations were associated with artemisinin resistance in different geographical areas in Southeast Asia, but here he presents that their role in artemisinin resistance is not known. I suggest that the authors should think around this and rephrase accordingly.

Response: Thanks for this observation however, this statements are not contradictory. A study by Miotto et al., 2011 confirmed the role of certain background mutations; fd-D193Y (ferredoxin gene), arps10-V127M (apicoplast ribosomal protein S10 gene), pfmdr2-T484I (multidrug resistance protein 2 gene), crt-I356T (chloroquine resistance transporter gene) and crt-N326S (chloroquine resistance transporter gene) in supporting the development of K13 mutations associated with artemisinin resistance. In our study we found only one mutation arsp10-V127M in one parasite sample collected in 2013 that had been previously reported in a study by Miotto et al. However, different background mutations from those reported by Miotto et al., were found in other genes and this are the mutations which the statement is referring to. This is because the role of the different background mutations that we found in our study is not known as they have not been confirmed/validated. This has been clarified in the revised manuscript. 

Reviewer #2: Reviewer’s comments: Line 298-300# - The author should support the statement with statistics.

Response: Thanks, we have performed statistical analysis and provided the results to support the statement. 

Reviewer #2: Reviewer’s comments: Line 295-300#- The author should fit in one or two implications of his findings in the conclusion to make a bit stronger.

Response: Thanks, this has been effected in the revised manuscript.

---

## [Decision Letter · Decision Letter 1]

31 Jan 2022

PONE-D-21-33254R1Prevalence of arps10, fd, pfmdr-2, pfcrt and pfkelch13 gene mutations in Plasmodium falciparum parasite population in UgandaPLOS ONE

Dear Dr. Ocan,

Thank you for submitting your manuscript to PLoS ONE. After careful consideration, we feel that your manuscript will likely be suitable for publication if the authors revise it to address specific points raised now by the reviewer. According to the reviewer, there are some specific areas where further improvements would be of substantial benefit to the readers.   For your guidance, a copy of the reviewers' comments was included below. 

Please submit your revised manuscript by Mar 17 2022 11:59PM If you will need more time than this to complete your revisions, please reply to this message or contact the journal office at plosone@plos.org. Please include the following items when submitting your revised manuscript:A rebuttal letter that responds to each point raised by the academic editor and reviewer(s). You should upload this letter as a separate file labeled 'Response to Reviewers'.A marked-up copy of your manuscript that highlights changes made to the original version. You should upload this as a separate file labeled 'Revised Manuscript with Track Changes'.An unmarked version of your revised paper without tracked changes. You should upload this as a separate file labeled 'Manuscript'.

We look forward to receiving your revised manuscript.

Kind regards,

Luzia Helena Carvalho, Ph.D.

Academic Editor

PLOS ONE

Reviewers' comments:

Reviewer's Responses to Questions

**Comments to the Author**

1. If the authors have adequately addressed your comments raised in a previous round of review and you feel that this manuscript is now acceptable for publication, you may indicate that here to bypass the “Comments to the Author” section, enter your conflict of interest statement in the “Confidential to Editor” section, and submit your "Accept" recommendation.

Reviewer #1: (No Response)

2. Is the manuscript technically sound, and do the data support the conclusions?

Reviewer #1: Partly

3. Has the statistical analysis been performed appropriately and rigorously? 

Reviewer #1: No

4. Have the authors made all data underlying the findings in their manuscript fully available?

Reviewer #1: Yes

5. Is the manuscript presented in an intelligible fashion and written in standard English?

Reviewer #1: Yes

6. Review Comments to the Author

Reviewer #1: The manuscript has been substantially improved since last revision. I believe the data presented in the paper are important and worthy of publication, but the paper still requires some work.

Below are some comments, which I believe still need to be addressed.

Major comments

(Lines numbers are based on numbers in the pdf version containing track changes.)

1). Authors have now added Table 4 to the Results section that shows the breakdown in prevalence of mutations in years 2005 and 2013, which is appreciated. However, there are errors in calculations for the 2005 and 2013 prevalence for all molecular markers. In this table the overall frequency of SNPs (for both years 2005 and 2013) is calculated correctly as total number of SNPs divided by total number of samples analysed for both years. However, when the authors calculate the prevalence in years 2005 or 2013, separately, the authors also use the total number of samples in the denominator, rather than using the number of samples from the respective year. This is not correct.

For an example, for SNP prevalence in K13 gene the total number of samples in which K13 gene was successfully sequenced was 142, of which 51 samples were collected in 2005 and 91 were collected in 2013. For overall (2005 and 2013) prevalence of K13 SNPs authors use the following calculation:

[6 SNPs)/142]*100%=4.2%, which is correct. However, for the year 2005 the prevalence is calculated as (2/142)* 100%=1.4% and for 2013 as (4/142)* 100%=2.8%, which is not correct. It should be calculated as (2/51)*100%= 3.9% and (4/91)*100%=4.4% for 2013, respectively. The same considerations and corrections would apply to other molecular markers analysed in the study (pfmdr-2 and arps10). Consequently, the contingency analysis for each SNP for 2005 and 2013 should be done with P-values calculated using exact Fisher’s test with 95% confidence intervals also recalculated for all markers/SNPs. This would also may change the conclusions reached in the paper and mentioned in the abstract.

I also suggest to include the number of samples analysed for every molecular marker gene for each year in the table 4 for ease of understanding as shown below:

Characteristic Molecular Marker/SNPs Collection Year Prevalence, % (n/N) P-value 95% CI

Pfkelch13 Overall 4.2 (6/142)

2005 1.4 3.9 (2/51)

2013 2.8 4.3 (4/91)

2). Lines: 37 (abstract), 128, 232, 236, 241, 280, 283, 285, 294. In these lines there are still references to 2015, which is incorrect as authors have confirmed that it should be 2013, the year when samples were collected. Could authors please check this for consistency throughout the text.

3). Lines 233-239: When describing the SNPs in Pfkelch13 gene, authors should refer to Table 3 initially, and then to Table 4 when describing numbers and prevalence in years 2005 and 2013. I also could not find Tables 1 and 2 in the main Text. Could authors please check and change the Table numbers if this is the case.

4). Authors might want to consider changing the titles for the Table 3 “Single nucleotide polymorphisms identified in pfkelch13 gene in samples collected in Kampala and Jinja, Uganda in 2005 and 2013.” Also, in footnotes in Table 3 it would be good to add the year when polymorphism was observed in those samples.

Also, a suggestion for Table 4 title would be “Prevalence of single nucleotide polymorphisms in pfkelch13, Pfmdr-2 and Pfarps 10 genes in Plasmodium falciparum samples collected in Kampala and Jinja, Uganda in 2005 and 2013”, instead of “Table 4: Distribution of the P. falciparum parasite mutations by year of sample collection, 2005 and 2013 (K13, n=142; pfmdr-2, n =186; arps10, n = 74)”.

I suggest to remove the word “different” from description of SNPs in pfmdr-2 and arps 10 genes, because it includes the SNPs at 484 and 127 in those genes reported by Miotto et al., 2015. The explanation in the text would be sufficient.

5) Abstract:

“In Uganda, Artemether-Lumefantrine and Artesunate are recommended for uncomplicated and 31 severe malaria respectively, but are currently threatened by parasite resistance”-I think the introductory sentence should be reworded as at present the data do not support resistance threat for this combination in Africa.

Also, suggest to replace “transcriptional” with “epigenetic” factors, implying changes in transcription.

In the abstract the actual SNPs, which were analysed in the present study¬ in the molecular markers genes should be stated.

The sentence “Overall, prevalence of SNPs in pfmdr-2 gene was 39.8% (74/186), of this 2.2% (4/186), 37.6% (70/186) occurred in 2005 and 2013 samples respectively” is misleading as it only refers to 2 “other” SNPs Y423F and I492V (not mentioned as Southeast Asian background mutations), but not to the SNP at pfmdr-2 T424I SNP reported by (Miotto et al., 2015). This need to be clarified in the abstract.

The prevalence numbers for all SNPs need to be recalculated with the number of samples analysed in the respective year used in the denominator. These may change the conclusion “There were more pfkelch13, arps10 and pfmdr-2 SNPs gene mutations in P. falciparum parasites collected a decade (2015) after introduction of Artemisinin combination therapies in malaria treatment.”

6). I believe that Discussion needs to be amended to reflect the following points:

1) The majority of samples before and after introduction of the ACTs are the wild type with only few SNPs identified in a very low fraction of samples (needs to be recalculated and presented). Importantly, they were different SNPs detected after introduction of ACTs, compared to those detected prior to, albeit all occurred at very low frequencies before and after introduction of the ACT.

2) As the K13 SNPs detected before and after introduction of ACTs are different it has to be stressed when making a comparison of K13 SNPs and conclusion that we see “There were more pfkelch13, arps10 and pfmdr-2 SNPs gene mutations in P. falciparum parasites collected a decade (2015) after introduction of Artemisinin combination therapies in 64 malaria treatment”. (lines 62-64) and in the discussion (Lines 320-322): “Additionally, our study found a general increase in the prevalence of K13 SNPs in samples collected after (2013) introduction of ACTs in malaria treatment in Uganda. It is likely that statistical analysis of the data would support this conclusion.

3) Importantly, there were no SNPs that were previously reported in Uganda “Balikagala et al., (34) demonstrated delayed artemisinin clearance among parasites carrying 675V and 469Y mutations in Uganda” (line 328-329) found in this study. These findings and their implications for the efficacy of the ACTs and potential for the resistance should be discussed further.

4) Absence of mutations in pfmdr-2 (T484I) and arps10 (V127M) in Africa was previously noted by Miotto et al 2015, with conclusion that these background mutations are specific to Southeast Asia. These findings have been subsequently confirmed by other recent studies that can be added to the discussion (Diakité, Seidina A S et al. “A comprehensive analysis of drug resistance molecular markers and Plasmodium falciparum genetic diversity in two malaria endemic sites in Mali.” Malaria journal vol. 18,1 361. 12 Nov. 2019, doi:10.1186/s12936-019-2986-5). Also findings on pfmdr-2 SNPs from Hussien M at al., Antimalarial drug resistance molecular makers of Plasmodium falciparum isolates from Sudan during 2015-2017. PLoS One. 2020 Aug 20;15(8)) should be discussed.

5) There were no mutations in Pfcrt N326S observed in the present study, whereas in Hussian et al there were 46.3% 96/207 samples carrying this SNP. This needs to be discussed.

6) Discussion on the importance in rise of other mutations in pfmdr-2 I492V 18.8? (35/?) 12.0 -39.0 Y423F 18.8 ?(35/?) needs to be added as these SNPs are the only ones that significantly more prevalent in 2013 (needs to be supported by stat. analysis).

Minor comments

Line 192: abbreviation PCR can be used as it has been previously defined. The same can be said about using SNP abbreviation throughout the text after it was first defined.

Lines 198: “double checked based on the 3D7 genome…” replace were tested

Line 234: (4.2%, 6/142) K13 single nucleotide polymorphisms (SNPs) were detected (Table 4). Of these, five (5) were non-synonymous and one (1) was a synonymous K13 mutation, (I587I).

Lines 235-236: four (54) (error), one (1) K13 non-synonymous mutations in the P. falciparum parasite samples collected 236 in 2015 and 2005 respectively.

In some sentences throughout the text numbers are duplicated in brackets in others are not. Please use consistent convention in the paper. Also, when listing SNPs it would be better to use their aa position numbers in the ascending order. For an example,

Line 238-241: One (1) K13 non-synonymous SNP Y588N occurred in a P. falciparum parasite sample collected in 2005. Five (5) K13 non-synonymous SNPs A578S, E596V, S600C and E643K and were detected in P. falciparum parasite samples collected in 2015.

Also, please use the same order (ascending) in the Table 4.

Line 276-278: Polymorphisms in the fd, pfmdr -2 and pfcrt genes that are markers of a genetic background where pfkelch13 mutations associated with slow artemisinin parasite clearance are likely to arise previously reported in Southeast Asia (Miotto, 20160 were not found in our study

Line 277: Pf mdr – 2 replace with pfmdr-2. Also, in some instances authors use pfkelch 13 and others K13 gene name. Please, use the same name throughout.

7. PLOS authors have the option to publish the peer review history of their article (what does this mean?). If published, this will include your full peer review and any attached files.

Reviewer #1: No

---

## [Author Response · Author response to Decision Letter 1]

9 Feb 2022

RESPONSE TO REVIEWER’S COMMENTS ON MANUSCRIPT PONE-D-21-33254R1

We thank the reviewers for the comments on our manuscript, addressing this comments have improved the manuscript. Here below is the summary of our responses to the comments raised by the reviewers.

Comment Reviewer #1: The manuscript has been substantially improved since last revision. I believe the data presented in the paper are important and worthy of publication, but the paper still requires some work.

Below are some comments, which I believe still need to be addressed.

Major comments

(Lines numbers are based on numbers in the pdf version containing track changes.)

Response: Thanks for the observation, this has been addressed in the revised manuscript

Comment Reviewer #1: 1). Authors have now added Table 4 to the Results section that shows the breakdown in prevalence of mutations in years 2005 and 2013, which is appreciated. However, there are errors in calculations for the 2005 and 2013 prevalence for all molecular markers. In this table the overall frequency of SNPs (for both years 2005 and 2013) is calculated correctly as total number of SNPs divided by total number of samples analysed for both years. However, when the authors calculate the prevalence in years 2005 or 2013, separately, the authors also use the total number of samples in the denominator, rather than using the number of samples from the respective year. This is not correct.

For an example, for SNP prevalence in K13 gene the total number of samples in which K13 gene was successfully sequenced was 142, of which 51 samples were collected in 2005 and 91 were collected in 2013. For overall (2005 and 2013) prevalence of K13 SNPs authors use the following calculation:

[6 SNPs)/142]*100%=4.2%, which is correct. However, for the year 2005 the prevalence is calculated as (2/142)* 100%=1.4% and for 2013 as (4/142)* 100%=2.8%, which is not correct. It should be calculated as (2/51)*100%= 3.9% and (4/91)*100%=4.4% for 2013, respectively. The same considerations and corrections would apply to other molecular markers analysed in the study (pfmdr-2 and arps10). Consequently, the contingency analysis for each SNP for 2005 and 2013 should be done with P-values calculated using exact Fisher’s test with 95% confidence intervals also recalculated for all markers/SNPs. This would also may change the conclusions reached in the paper and mentioned in the abstract.

I also suggest to include the number of samples analysed for every molecular marker gene for each year in the table 4 for ease of understanding as shown below:

Characteristic Molecular Marker/SNPs Collection Year Prevalence, % (n/N) P-value 95% CI

Pfkelch13 Overall 4.2 (6/142)

2005 1.4 3.9 (2/51)

2013 2.8 4.3 (4/91)

Response: Thanks for the comments, the re-analysis has been done using Fisher’s Exact test and the results are presented in Table 4. We have included the requested numbers in the revised table. 

Comment Reviewer #1: 2). Lines: 37 (abstract), 128, 232, 236, 241, 280, 283, 285, 294. In these lines there are still references to 2015, which is incorrect as authors have confirmed that it should be 2013, the year when samples were collected. Could authors please check this for consistency throughout the text.

Response: Thanks for the observation. We have corrected the year to 2013 all through the revised manuscript. 

Comment Reviewer #1: 3). Lines 233-239: When describing the SNPs in Pfkelch13 gene, authors should refer to Table 3 initially, and then to Table 4 when describing numbers and prevalence in years 2005 and 2013. I also could not find Tables 1 and 2 in the main Text. Could authors please check and change the Table numbers if this is the case.

Response: Tables 1 and 2 are referenced in the main text under the methods section but provided as additional material. Revision has been made on reference to Table 3 first when describing the Pfkelch13 gene.

Comment Reviewer #1: 4). Authors might want to consider changing the titles for the Table 3 “Single nucleotide polymorphisms identified in pfkelch13 gene in samples collected in Kampala and Jinja, Uganda in 2005 and 2013.” Also, in footnotes in Table 3 it would be good to add the year when polymorphism was observed in those samples.

Also, a suggestion for Table 4 title would be “Prevalence of single nucleotide polymorphisms in pfkelch13, Pfmdr-2 and Pfarps 10 genes in Plasmodium falciparum samples collected in Kampala and Jinja, Uganda in 2005 and 2013”, instead of “Table 4: Distribution of the P. falciparum parasite mutations by year of sample collection, 2005 and 2013 (K13, n=142; pfmdr-2, n =186; arps10, n = 74)”.

I suggest to remove the word “different” from description of SNPs in pfmdr-2 and arps 10 genes, because it includes the SNPs at 484 and 127 in those genes reported by Miotto et al., 2015. The explanation in the text would be sufficient.

Response: This correction has been effected in the revised manuscript

Comment Reviewer #1: 5) Abstract:

“In Uganda, Artemether-Lumefantrine and Artesunate are recommended for uncomplicated and 31 severe malaria respectively, but are currently threatened by parasite resistance”-I think the introductory sentence should be reworded as at present the data do not support resistance threat for this combination in Africa.

Also, suggest to replace “transcriptional” with “epigenetic” factors, implying changes in transcription.

In the abstract the actual SNPs, which were analysed in the present study¬ in the molecular markers genes should be stated.

The sentence “Overall, prevalence of SNPs in pfmdr-2 gene was 39.8% (74/186), of this 2.2% (4/186), 37.6% (70/186) occurred in 2005 and 2013 samples respectively” is misleading as it only refers to 2 “other” SNPs Y423F and I492V (not mentioned as Southeast Asian background mutations), but not to the SNP at pfmdr-2 T424I SNP reported by (Miotto et al., 2015). This need to be clarified in the abstract.

The prevalence numbers for all SNPs need to be recalculated with the number of samples analysed in the respective year used in the denominator. These may change the conclusion “There were more pfkelch13, arps10 and pfmdr-2 SNPs gene mutations in P. falciparum parasites collected a decade (2015) after introduction of Artemisinin combination therapies in malaria treatment.”

Response: The introductory sentence is appropriate and current data does support resistance threat to artemisinin agents in Africa as shown in a study by Balikagala et al., 2021. This sentence has been maintained in the revised manuscript. 

The word ‘transcriptional’ has been replaced with ‘epigenetic’ in the revised manuscript.

The sentence “Overall, prevalence of SNPs in pfmdr-2 gene was 39.8% (74/186), of this 2.2% (4/186), 37.6% (70/186) occurred in 2005 and 2013 samples respectively is appropriate. However, we have removed the word ‘overall’ in the revised manuscript. 

The prevalence of all the SNPs has been recalculated however, the direction of the findings remained the same and thus the conclusion “There were more pfkelch13, arps10 and pfmdr-2 SNPs gene mutations in P. falciparum parasites collected a decade (2015) after introduction of Artemisinin combination therapies in malaria treatment.” Has been maintained in the revised manuscript. 

Comment Reviewer #1: 6). I believe that Discussion needs to be amended to reflect the following points:

1) The majority of samples before and after introduction of the ACTs are the wild type with only few SNPs identified in a very low fraction of samples (needs to be recalculated and presented). Importantly, they were different SNPs detected after introduction of ACTs, compared to those detected prior to, albeit all occurred at very low frequencies before and after introduction of the ACT.

2) As the K13 SNPs detected before and after introduction of ACTs are different it has to be stressed when making a comparison of K13 SNPs and conclusion that we see “There were more pfkelch13, arps10 and pfmdr-2 SNPs gene mutations in P. falciparum parasites collected a decade (2015) after introduction of Artemisinin combination therapies in 64 malaria treatment”. (lines 62-64) and in the discussion (Lines 320-322): “Additionally, our study found a general increase in the prevalence of K13 SNPs in samples collected after (2013) introduction of ACTs in malaria treatment in Uganda. It is likely that statistical analysis of the data would support this conclusion.

3) Importantly, there were no SNPs that were previously reported in Uganda “Balikagala et al., (34) demonstrated delayed artemisinin clearance among parasites carrying 675V and 469Y mutations in Uganda” (line 328-329) found in this study. These findings and their implications for the efficacy of the ACTs and potential for the resistance should be discussed further.

4) Absence of mutations in pfmdr-2 (T484I) and arps10 (V127M) in Africa was previously noted by Miotto et al 2015, with conclusion that these background mutations are specific to Southeast Asia. These findings have been subsequently confirmed by other recent studies that can be added to the discussion (Diakité, Seidina A S et al. “A comprehensive analysis of drug resistance molecular markers and Plasmodium falciparum genetic diversity in two malaria endemic sites in Mali.” Malaria journal vol. 18,1 361. 12 Nov. 2019, doi:10.1186/s12936-019-2986-5). Also findings on pfmdr-2 SNPs from Hussien M at al., Antimalarial drug resistance molecular makers of Plasmodium falciparum isolates from Sudan during 2015-2017. PLoS One. 2020 Aug 20;15(8)) should be discussed.

5) There were no mutations in Pfcrt N326S observed in the present study, whereas in Hussian et al there were 46.3% 96/207 samples carrying this SNP. This needs to be discussed.

6) Discussion on the importance in rise of other mutations in pfmdr-2 I492V 18.8? (35/?) 12.0 -39.0 Y423F 18.8 ?(35/?) needs to be added as these SNPs are the only ones that significantly more prevalent in 2013 (needs to be supported by stat. analysis).

Response: Thanks for the comment, the discussion has been amended to reflect the suggestions. The role the different background mutations identified from the genes need to be validated among the African Plasmodium falciparum parasites. 

Minor comments

Comment Reviewer #1: Line 192: abbreviation PCR can be used as it has been previously defined. The same can be said about using SNP abbreviation throughout the text after it was first defined.

Response: Thanks, this has been corrected in the revised manuscript

Comment Reviewer #1: Lines 198: “double checked based on the 3D7 genome…” replace were tested

Response: Thanks, this has been revised in the corrected manuscript

Comment Reviewer #1: Line 234: (4.2%, 6/142) K13 single nucleotide polymorphisms (SNPs) were detected (Table 4). Of these, five (5) were non-synonymous and one (1) was a synonymous K13 mutation, (I587I).

Response: This has been corrected in the revised manuscript. 

Comment Reviewer #1: Lines 235-236: four (54) (error), one (1) K13 non-synonymous mutations in the P. falciparum parasite samples collected 236 in 2015 and 2005 respectively.

In some sentences throughout the text numbers are duplicated in brackets in others are not. Please use consistent convention in the paper. Also, when listing SNPs it would be better to use their aa position numbers in the ascending order. For an example,

Response: Thanks for the observation, this has been corrected in the revised manuscript. 

Comment Reviewer #1: Line 238-241: One (1) K13 non-synonymous SNP Y588N occurred in a P. falciparum parasite sample collected in 2005. Five (5) K13 non-synonymous SNPs A578S, E596V, S600C and E643K and were detected in P. falciparum parasite samples collected in 2015.

Also, please use the same order (ascending) in the Table 4.

Response: Thanks for the comment, this has been corrected in the revised manuscript. 

Comment Reviewer #1: Line 276-278: Polymorphisms in the fd, pfmdr -2 and pfcrt genes that are markers of a genetic background where pfkelch13 mutations associated with slow artemisinin parasite clearance are likely to arise previously reported in Southeast Asia (Miotto, 20160 were not found in our study

Response: Thanks for the observation, the citation, Miotto et al., has been included into the sentence in the revised manuscript. 

Comment Reviewer #1: Line 277: Pf mdr – 2 replace with pfmdr-2. Also, in some instances authors use pfkelch 13 and others K13 gene name. Please, use the same name throughout.

Response: Thanks for the comment, this has been corrected in the revised manuscript.

---

## [Decision Letter · Decision Letter 2]

14 Mar 2022

PONE-D-21-33254R2Prevalence of arps10, fd, pfmdr-2, pfcrt and pfkelch13 gene mutations in Plasmodium falciparum parasite population in UgandaPLOS ONE

Dear Dr. Ocan,

Thank you for resubmitting your manuscript for review to PLoS ONE. After careful consideration, we feel that your manuscript will likely be suitable for publication if it is revised to address  major  points raised by the reviewer. While the subject of the MS was of interest, the authors did not properly address relevant topics raised during the peer review process.At this time, we strongly recommend that the authors include the modifications requested by the reviewer. For your guidance, a copy of the reviewers' comments was included below.  

We look forward to receiving your revised manuscript.

Kind regards,

Luzia Helena Carvalho, Ph.D.

Academic Editor

PLOS ONE

Reviewers' comments:

Reviewer's Responses to Questions

**Comments to the Author**

1. If the authors have adequately addressed your comments raised in a previous round of review and you feel that this manuscript is now acceptable for publication, you may indicate that here to bypass the “Comments to the Author” section, enter your conflict of interest statement in the “Confidential to Editor” section, and submit your "Accept" recommendation.

Reviewer #1: (No Response)

2. Is the manuscript technically sound, and do the data support the conclusions?

Reviewer #1: Yes

3. Has the statistical analysis been performed appropriately and rigorously? 

Reviewer #1: No

4. Have the authors made all data underlying the findings in their manuscript fully available?

Reviewer #1: Yes

5. Is the manuscript presented in an intelligible fashion and written in standard English?

Reviewer #1: Yes

6. Review Comments to the Author

Reviewer #1: The authors have partially addressed some of my previously mentioned points. However, I believe that the paper still requires further improvements and authors should address the following points raised in my previous comments (please see below).

1). The Abstract/Results/Discussion (as previously mentioned)

The aim of the paper was stated in the Introduction (lines 113-116) as “We therefore, intended in this study to assess and 114 compare the prevalence of fd-D193Y, aps10-V127M, pfmdr2-T484I, crt-I356T, crt-N326S and 115 K13 mutations in P. falciparum parasites before (2005) and a decade (2013) after introduction of 116 artemisinin-based combination therapies for malaria treatment in Uganda (18)”. Therefore, the SNPs prevalences in those years should be calculated correctly as previously mentioned.

In the present revision authors have provided the break-down in numbers of samples analyzed in 2005 and 2013 and calculated the respective prevalence for each collection time, which are now shown in Table 4. However, the authors have not amended or changed the data in the Abstract and in the Results sections for the prevalence values for 2005 or 2013 to reflect that, and still used the total number of samples analysed for both years overall in the denominator. As the primary aim of this paper is to compare the prevalence of the SNPs in samples collected in 2005 and in 2013 (after introduction of ACTs), the respective prevalence should be calculated correctly. When reporting (throughout the paper), the prevalence values should be accompanied by the fraction (n/N) showing the number of samples with SNPs (n) and N number of samples analysed in this time period, as well as 95% confidence interval (CI). Also, when comparing these values, the authors should state if there is (or not) a statistically significant difference between the prevalence values in 2005 and 2013 (with P-value), which are calculated based on these numbers.

Also, if authors could use the prevalence values in the Discussion rather than just mentioning a particular detected SNP, so the readers could put it in a context and appreciate the significance of the findings.

Please see below the specific examples (line numbers are as in the “Clean” version).

Lines 47-55 (Abstract) and throughout the Results section authors should amend the abstract and the result with the SNP prevalence data for 2005 and 2013 years respectively, where the number of SNPs is divided by the number of samples analysed in that particular time period (i.e. either in 2005 or in 2013), not the combined overall number of samples to match the “Prevalence” data shown in Table 4.

2). I suggest Tables 1 and 2 should be renamed S1 and S2 as they are in the Supplementary material not in the main text and renumber Tables 3 and 4.

3). Table 4. Please use consistent gene names nomenclature in the Table title and in the Table (i.e. Pfkelch 13 or K13).

Column 2 and 3 have (%) in the columns’ headings but don’t actually show these percentages. These columns can be combined with the column “Prevalence, % (n)” if the number of samples (N) analysed per actual time period can be added “Prevalence, % (n/N)” as previously suggested.

4). Results section.

Lines 269 The sentence about the SNPs in pfmdr2 (“There was a significant relationship in the prevalence of mutations in the pfmdr-2 gene….”) should be included in the next section, as this section is about SNPs in K13 gene. Also, could authors state if there is a statistically significant difference between the prevalence values instead of "significant relationship")

Throughout the Results section the actual prevalence values from Table 4 should be used with confidence intervals and P-values for the prevalence values for 2005 and 2013, respectively. It is very important as it allows readers to appreciate the statistical analysis of the data. Also, if authors also could list codons in the ascending order when mentioning SNPs throughout the text (for example in line 278).

Discussion

If authors could use the prevalence values and sample numbers throughout the Discussion as well when commenting on identified SNPs and also compare with those in the referenced published literature.

As the aim of the paper was to detect if the background mutations identified by Miotto et al.in SEA are present in Uganda, it would be more logical to start the Discussion with the findings related to this primary aim and then continue discussing the secondary (also important) findings. I would suggest to rearrange the paragraphs in the Discussion and start the discussion with the text shown in lines 309-319.

319: “Plasmodium falciparum artemisinin resistance is a heritable trait with a genetic basis (7). Genome modification studies have shown that the impact of various K13 pfkelch13 mutations on P. falciparum artemisinin clearance and survival rates of ring stage parasites is dependent on the genetic background (27). The risk of emergence of K13pfkelch13 mutations associated with delayed artemisinin parasite clearance is thus driven by specific parasite genetic background (8, 314 28). In our current study polymorphisms in the fd, pfmdr-2 and pfcrt genes which support the rise of K13 mutations associated with delayed artemisinin clearance of P. falciparum parasites reported in Southeast Asia (8) were not found. However, SNPs in the pfmdr-2, pfcrt, and arps10 genes not reported in a study by Miotto et al.,(8) and other previous studies in Africa (29) were detected in our current study. There is need to validate the role the identified background mutations play in artemisinin resistance development among African P. falciparum parasites.”

Lines 299-302: In the following sentences it is not clear what is actually “similar”: “A recent study in South Sudan (26) which analyzed samples collected in 2015-2017 after introduction of artemisinin agents in malaria treatment detected genetic background mutation in Pfcrt N326S gene which was previously reported in Southeast Asia and associated with artemisinin resistance. This is similar to a finding from our current study where we detected a background mutation, V127M in the arps10 gene that has been shown to support development of K13 mutations associated with artemisinin resistance in Southeast Asia(8)”

Add the following “While we did not find Pfcrt N326S SNP in our samples, we detected…”

“… after introduction of artemisinin agents in malaria treatment detected genetic background mutation in Pfcrt N326S gene which was previously reported in Southeast Asia and associated with artemisinin resistance. While we did not find Pfcrt N326S SNP in our samples, we detected a background mutation, V127M in the arps10 gene that has been also shown to support development of K13 mutations associated with artemisinin resistance in Southeast Asia(8)”.

Line 320: This statement is ambiguous: “We detected different K13 SNPs in P. falciparum samples collected both prior (2005) and after (2013) introduction of artemisinin combination therapies in malaria treatment in Uganda.

Suggest to change the sentence to indicate that K13 SNPs identified in the present study are different to those previously implicated in artemisinin resistance in Africa and SEA. For example, “In our samples collected in 2005 and 2013 we detected K13 SNPs different to those previously implicated in artemisinin resistance (ref). or "We detected different K13 SNPs in P. falciparum samples collected in 2005 compared to those collected in 2013 after introduction of artemisinin combination therapies in malaria treatment in Uganda".

Line 322: Our findings are similar to a recent study in Uganda that reported K13 mutations in the P. falciparum 323 parasites (30)- Needs clarification similar in what way?

Line 323-324. “Additionally, our study found a general increase in the prevalence of K13 SNPs in samples collected after (2013) introduction of ACTs in malaria treatment in Uganda”.

This statement is bit misleading since the is no statistically significant differences in the prevalences of these SNPs before and after introduction of ACTs.

The similar statement in Conclusions (below) can also be misinterpreted with the vagueness of the term “generally”. While it is correct in case of novel SNPs in pfmdr2, differences in SNPs for K13 and arsp10 were not statistically significant. Conclusions, should also reflect the findings related to the primary aim of the study and state that there were neither K13 SNPs previously implicated in artemisinin resistance in Africa and SEA, nor mutations in the associated background genes identified in this study with exception of one SNP in arps10 gene.

Line 331 “The proportions of K13, arsp10, and pfmdr-2 gene mutations were generally higher in P. falciparum parasites collected after introduction of artemisinin combination therapies in malaria treatment in Uganda

Other comments.

Line 34: Spaces are needed between P. and falciparum

Lines 36,56, 115: Saying it is “a decade” between 2005 and 2013 is not accurate-it is 8 years.

Line 257 fcrt replace for pfcrt

7. PLOS authors have the option to publish the peer review history of their article (what does this mean?). If published, this will include your full peer review and any attached files.

Reviewer #1: No

---

## [Author Response · Author response to Decision Letter 2]

30 Mar 2022

RESPONSE TO REVIEWER’S COMMENTS ON MANUSCRIPT PONE-D-21-33254R2

We are grateful to the reviewers for the comments raised on our manuscript. The comments have been addressed and here below is a summary of the point-by-point response to the different questions. 

Comment, Reviewer #1: The authors have partially addressed some of my previously mentioned points. However, I believe that the paper still requires further improvements and authors should address the following points raised in my previous comments (please see below).

1). The Abstract/Results/Discussion (as previously mentioned)

The aim of the paper was stated in the Introduction (lines 113-116) as “We therefore, intended in this study to assess and 114 compare the prevalence of fd-D193Y, aps10-V127M, pfmdr2-T484I, crt-I356T, crt-N326S and 115 K13 mutations in P. falciparum parasites before (2005) and a decade (2013) after introduction of 116 artemisinin-based combination therapies for malaria treatment in Uganda (18)”. Therefore, the SNPs prevalences in those years should be calculated correctly as previously mentioned.

Response:

Thanks for this observation, actually the calculations were made as previously requested except that we forgot to remove the denominator ‘142’ and place the correct denominator used in the analysis, 51 (for 2005) and 91 (for 2013) for K13 mutations. This error has been corrected in the revised manuscript. 

Comment: In the present revision authors have provided the break-down in numbers of samples analyzed in 2005 and 2013 and calculated the respective prevalence for each collection time, which are now shown in Table 4. However, the authors have not amended or changed the data in the Abstract and in the Results sections for the prevalence values for 2005 or 2013 to reflect that, and still used the total number of samples analysed for both years overall in the denominator. 

Response: 

Thanks for this observation, this was an editorial error as we forgot to adjust the denominators in the revised manuscript despite the fact that the analysis were done following the adjusted denominators as previously guided. We have adjusted the analysis as requested in the revised manuscript. 

Comment: As the primary aim of this paper is to compare the prevalence of the SNPs in samples collected in 2005 and in 2013 (after introduction of ACTs), the respective prevalence should be calculated correctly. When reporting (throughout the paper), the prevalence values should be accompanied by the fraction (n/N) showing the number of samples with SNPs (n) and N number of samples analysed in this time period, as well as 95% confidence interval (CI). Also, when comparing these values, the authors should state if there is (or not) a statistically significant difference between the prevalence values in 2005 and 2013 (with P-value), which are calculated based on these numbers.

Response:

Thanks for the comment, this analysis are clearly stated in table 4. We have further provided the descriptive summaries for the different mutations in the results section of the revised manuscript. 

Comment: Also, if authors could use the prevalence values in the Discussion rather than just mentioning a particular detected SNP, so the readers could put it in a context and appreciate the significance of the findings.

Response:

Thanks for the comment, this has been incorporated in the revised manuscript.

Comment: Please see below the specific examples (line numbers are as in the “Clean” version).

Response: This is noted

Comment: Lines 47-55 (Abstract) and throughout the Results section authors should amend the abstract and the result with the SNP prevalence data for 2005 and 2013 years respectively, where the number of SNPs is divided by the number of samples analysed in that particular time period (i.e. either in 2005 or in 2013), not the combined overall number of samples to match the “Prevalence” data shown in Table 4.

Response: Thanks, this has been effected in the revised manuscript

Comment: 2). I suggest Tables 1 and 2 should be renamed S1 and S2 as they are in the Supplementary material not in the main text and renumber Tables 3 and 4.

Response: The information in the tables S1 and S2 in the supplementary material is different from the information in tables 1 and 2 in the main text. We therefore, think the current numbering is appropriate and has been maintained in the revised manuscript.

3). Table 4. Please use consistent gene names nomenclature in the Table title and in the Table (i.e. Pfkelch 13 or K13).

Response: This has been noted and corrected in the revised manuscript

Comment: Column 2 and 3 have (%) in the columns’ headings but don’t actually show these percentages. These columns can be combined with the column “Prevalence, % (n)” if the number of samples (N) analysed per actual time period can be added “Prevalence, % (n/N)” as previously suggested.

Response: This has been noted however, from your previous comment you requested for a clear separation of number of samples which had and those which did not have mutations and we effected. We find it confusing to now request for merging the numbers again. We have however, taken note of the comment and made adjustments in table 4 as indicated in the revised manuscript. 

4). Results section.

Comment: Lines 269 The sentence about the SNPs in pfmdr2 (“There was a significant relationship in the prevalence of mutations in the pfmdr-2 gene….”) should be included in the next section, as this section is about SNPs in K13 gene. Also, could authors state if there is a statistically significant difference between the prevalence values instead of "significant relationship")

Response;

This is noted and has been effected in the revised manuscript

Comments: Throughout the Results section the actual prevalence values from Table 4 should be used with confidence intervals and P-values for the prevalence values for 2005 and 2013, respectively. It is very important as it allows readers to appreciate the statistical analysis of the data. Also, if authors also could list codons in the ascending order when mentioning SNPs throughout the text (for example in line 278).

Response: Thanks for the comment, this has been effected in the revised manuscript

Discussion

Comment: If authors could use the prevalence values and sample numbers throughout the Discussion as well when commenting on identified SNPs and also compare with those in the referenced published literature.

Response: Thanks however, we this as a repetition of the results which are clearly stated in the results section. We have however, effected the adjustment as requested. 

Comment: As the aim of the paper was to detect if the background mutations identified by Miotto et al.in SEA are present in Uganda, it would be more logical to start the Discussion with the findings related to this primary aim and then continue discussing the secondary (also important) findings. I would suggest to rearrange the paragraphs in the Discussion and start the discussion with the text shown in lines 309-319.

319: “Plasmodium falciparum artemisinin resistance is a heritable trait with a genetic basis (7). Genome modification studies have shown that the impact of various K13 pfkelch13 mutations on P. falciparum artemisinin clearance and survival rates of ring stage parasites is dependent on the genetic background (27). The risk of emergence of K13pfkelch13 mutations associated with delayed artemisinin parasite clearance is thus driven by specific parasite genetic background (8, 314 28). In our current study polymorphisms in the fd, pfmdr-2 and pfcrt genes which support the rise of K13 mutations associated with delayed artemisinin clearance of P. falciparum parasites reported in Southeast Asia (8) were not found. However, SNPs in the pfmdr-2, pfcrt, and arps10 genes not reported in a study by Miotto et al.,(8) and other previous studies in Africa (29) were detected in our current study. There is need to validate the role the identified background mutations play in artemisinin resistance development among African P. falciparum parasites.”

Response: This is noted and has been effected in the revised manuscript.

Comment: Lines 299-302: In the following sentences it is not clear what is actually “similar”: “A recent study in South Sudan (26) which analyzed samples collected in 2015-2017 after introduction of artemisinin agents in malaria treatment detected genetic background mutation in Pfcrt N326S gene which was previously reported in Southeast Asia and associated with artemisinin resistance. This is similar to a finding from our current study where we detected a background mutation, V127M in the arps10 gene that has been shown to support development of K13 mutations associated with artemisinin resistance in Southeast Asia(8)”

Add the following “While we did not find Pfcrt N326S SNP in our samples, we detected…”

“… after introduction of artemisinin agents in malaria treatment detected genetic background mutation in Pfcrt N326S gene which was previously reported in Southeast Asia and associated with artemisinin resistance. While we did not find Pfcrt N326S SNP in our samples, we detected a background mutation, V127M in the arps10 gene that has been also shown to support development of K13 mutations associated with artemisinin resistance in Southeast Asia(8)”.

Response: Thanks for the comment. This has been effected in the revised manuscript

Comment: Line 320: This statement is ambiguous: “We detected different K13 SNPs in P. falciparum samples collected both prior (2005) and after (2013) introduction of artemisinin combination therapies in malaria treatment in Uganda.

Suggest to change the sentence to indicate that K13 SNPs identified in the present study are different to those previously implicated in artemisinin resistance in Africa and SEA. For example, “In our samples collected in 2005 and 2013 we detected K13 SNPs different to those previously implicated in artemisinin resistance (ref). or "We detected different K13 SNPs in P. falciparum samples collected in 2005 compared to those collected in 2013 after introduction of artemisinin combination therapies in malaria treatment in Uganda".

Response: The comment is noted, we have made the suggested adjustments in the revised manuscript.

Comment: Line 322: Our findings are similar to a recent study in Uganda that reported K13 mutations in the P. falciparum 323 parasites (30)- Needs clarification similar in what way?

Response: Thanks for the comment, this sentence has be deleted from the revised manuscript. 

Comment: Line 323-324. “Additionally, our study found a general increase in the prevalence of K13 SNPs in samples collected after (2013) introduction of ACTs in malaria treatment in Uganda”.

This statement is bit misleading since the is no statistically significant differences in the prevalences of these SNPs before and after introduction of ACTs.

Response: The statement has been corrected in the revised manuscript. Although not statistically significant, there is an observed trend towards increase in the prevalence of K13 mutations following introduction of artemisinin agents in malaria treatment. 

Comment: The similar statement in Conclusions (below) can also be misinterpreted with the vagueness of the term “generally”. While it is correct in case of novel SNPs in pfmdr2, differences in SNPs for K13 and arsp10 were not statistically significant. Conclusions, should also reflect the findings related to the primary aim of the study and state that there were neither K13 SNPs previously implicated in artemisinin resistance in Africa and SEA, nor mutations in the associated background genes identified in this study with exception of one SNP in arps10 gene.

Response: This is noted and has been corrected in the revised manuscript

Comment: Line 331 “The proportions of K13, arsp10, and pfmdr-2 gene mutations were generally higher in P. falciparum parasites collected after introduction of artemisinin combination therapies in malaria treatment in Uganda

Response: This has been revised to refer to only pfmdr-2 gene mutation in the revised manuscript. 

Other comments.

Comment: Line 34: Spaces are needed between P. and falciparum

Response: This has been corrected

Comment: Lines 36,56, 115: Saying it is “a decade” between 2005 and 2013 is not accurate-it is 8 years.

Response: Thanks for the comment, this has been corrected in the revised manuscript

Comment: Line 257 fcrt replace for pfcrt

Response: This has been corrected in the revised manuscript.

---

## [Decision Letter · Decision Letter 3]

18 Apr 2022

PONE-D-21-33254R3Prevalence of arps10, fd, pfmdr-2, pfcrt and pfkelch13 gene mutations in Plasmodium falciparum parasite population in UgandaPLOS ONE

Dear Dr.  Ocan,

Thank you for submitting your manuscript for review to PLoS ONE. After careful consideration, we feel that your manuscript will likely be suitable for publication if the authors revise it to address critical points raised by the reviewer. According to reviewer, there are some specific areas where further improvements would be of substantial benefit to the readers. A copy of the reviewers’ comments was included for your information.

We look forward to receiving your revised manuscript.

Kind regards,

Luzia Helena Carvalho, Ph.D.

Academic Editor

PLOS ONE

Journal Requirements:

Reviewers' comments:

Reviewer's Responses to Questions

**Comments to the Author**

1. If the authors have adequately addressed your comments raised in a previous round of review and you feel that this manuscript is now acceptable for publication, you may indicate that here to bypass the “Comments to the Author” section, enter your conflict of interest statement in the “Confidential to Editor” section, and submit your "Accept" recommendation.

Reviewer #1: (No Response)

2. Is the manuscript technically sound, and do the data support the conclusions?

Reviewer #1: Yes

3. Has the statistical analysis been performed appropriately and rigorously? 

Reviewer #1: Yes

4. Have the authors made all data underlying the findings in their manuscript fully available?

Reviewer #1: Yes

5. Is the manuscript presented in an intelligible fashion and written in standard English?

Reviewer #1: Yes

6. Review Comments to the Author

Reviewer #1: Authors have addressed almost all of the comments. I believe that the MS can be accepted for publication after the following minor comments are addressed-please see below. The version with Track changes is used for line references.

Line 47: Authors inserted “…and arps10 (p=0.238)”, which is repetitive with the same statement below in lines 56-57 “…There was no statistically significant difference relationship (p=0.238) in the prevalence of arps10 SNPs..”. Suggest to delete “and arps10 (p=0.238)” in line 47.

Line 250: Table 2 in Column “Prevalence of mutation, %(n/N)” “N” is actually not shown, while it is stated in the column heading and in the footnote to the table. As I previously, suggested, N (the denominator) needs to be added for every number in this column consistent with the column’s heading. It would make it easier to comprehend the data and consistent with the format of data shown in the text.

Line 114: “South east Asia” please change for Southeast or South East Asia.

Line 141-142 “…Confirmation of the presence of P. falciparum parasites in the stored(?) blood samples was done by two techniques namely malaria rapid diagnostic test (mRDT) and microscopy”. Were RDT and microscopy done before samples were frozen or after samples were thawed after storage. While RDTs are possible to do on frozen samples I am not sure if the quality slides could be produced after the blood is lysed upon thawing. If authors could clarify this statement.

Line 205: “Correlation analysis was done using Fisher’s exact test to assess the relationship between mutations and year of sample collection” Suggest to change it to “.. using Fisher’s exact test to assess the differences in the prevalence of mutations in 2005 and 2013.”

Line 213 “…(4.2%, 6/142) K13 SNPs were detected (Table 2)”. Suggest to replace with “… SNPs were detected (Tables 1 and 2)” to reference the Tables in their order of appearance.

7. PLOS authors have the option to publish the peer review history of their article (what does this mean?). If published, this will include your full peer review and any attached files.

Reviewer #1: No

---

## [Author Response · Author response to Decision Letter 3]

20 Apr 2022

RESPONSE TO REVIEWER’S COMMENTS ON MANUSCRIPT PONE-D-21-33254R3

Comments, Reviewer #1: Authors have addressed almost all of the comments. I believe that the MS can be accepted for publication after the following minor comments are addressed-please see below. The version with Track changes is used for line references.

Response: Thanks for the comments that have helped improve the manuscript, we are grateful. 

Comment, Reviewer #1: Line 47: Authors inserted “…and arps10 (p=0.238)”, which is repetitive with the same statement below in lines 56-57 “…There was no statistically significant difference relationship (p=0.238) in the prevalence of arps10 SNPs..”. Suggest to delete “and arps10 (p=0.238)” in line 47.

Response: This has been effected in the revised manuscript

Comment, Reviewer #1: Line 250: Table 2 in Column “Prevalence of mutation, %(n/N)” “N” is actually not shown, while it is stated in the column heading and in the footnote to the table. As I previously, suggested, N (the denominator) needs to be added for every number in this column consistent with the column’s heading. It would make it easier to comprehend the data and consistent with the format of data shown in the text.

Response: Thanks, this has been provided as guided in the revised manuscript

Comment, Reviewer #1: Line 114: “South east Asia” please change for Southeast or South East Asia.

Response: Thanks, this has been corrected in the revised manuscript. 

Comment, Reviewer #1: Line 141-142 “…Confirmation of the presence of P. falciparum parasites in the stored(?) blood samples was done by two techniques namely malaria rapid diagnostic test (mRDT) and microscopy”. Were RDT and microscopy done before samples were frozen or after samples were thawed after storage. While RDTs are possible to do on frozen samples I am not sure if the quality slides could be produced after the blood is lysed upon thawing. If authors could clarify this statement.

Response: Sorry, we did not make it clear enough. Microscopy was performed in the field before the blood samples were frozen by the primary studies. In our current study, we conducted pfHRP-2 malaria rapid diagnosis (mRDT) to screen for frozen samples that had P. falciparum parasites. This has been re-written in the revised manuscript revised manuscript. 

Comment, Reviewer #1: Line 205: “Correlation analysis was done using Fisher’s exact test to assess the relationship between mutations and year of sample collection” Suggest to change it to “.. using Fisher’s exact test to assess the differences in the prevalence of mutations in 2005 and 2013.”

Response: Thanks, this has been corrected in the revised manuscript. 

Comment, Reviewer #1: Line 213 “…(4.2%, 6/142) K13 SNPs were detected (Table 2)”. Suggest to replace with “… SNPs were detected (Tables 1 and 2)” to reference the Tables in their order of appearance.

Response: Thanks for the comment, this has been corrected in the revised manuscript

---

## [Editor Report · Decision Letter 4]

22 Apr 2022

Prevalence of arps10, fd, pfmdr-2, pfcrt and pfkelch13 gene mutations in Plasmodium falciparum parasite population in Uganda

PONE-D-21-33254R4

Dear Dr. Ocan,

We’re pleased to inform you that your manuscript has been judged scientifically suitable for publication and will be formally accepted for publication once it meets all outstanding technical requirements.

Kind regards,

Luzia Helena Carvalho, Ph.D.

Academic Editor

PLOS ONE
---

## [Editor Report · Acceptance letter]

27 Apr 2022

PONE-D-21-33254R4 

Prevalence of *arps10, fd, pfmdr-2, pfcrt and pfkelch13* gene mutations in *Plasmodium falciparum* parasite population in Uganda 

Dear Dr. Ocan:

I'm pleased to inform you that your manuscript has been deemed suitable for publication in PLOS ONE. Congratulations! Your manuscript is now with our production department. 

Kind regards, 

on behalf of

Dr. Luzia Helena Carvalho 

Academic Editor

PLOS ONE